# The HEDGEHOG-GLI1 pathway is important for fibroproliferative properties in keloids and as a candidate therapeutic target

Mamiko Tosa[1,2,3], Yoshinori Abe [1,3], Seiko Egawa[1], Tomoka Hatakeyama[1], Chihiro Iwaguro[1], Ryotaro Mitsugi[1], Ayaka Moriyama[1], Takumi Sano[1], Rei Ogawa[2] & Nobuyuki Tanaka [1✉]

Keloids are benign fibroproliferative skin tumors caused by aberrant wound healing that can negatively impact patient quality of life. The lack of animal models has limited research on pathogenesis or developing effective treatments, and the etiology of keloids remains unknown. Here, we found that the characteristics of stem-like cells from keloid lesions and the surrounding dermis differ from those of normal skin. Furthermore, the HEDGEHOG (HH) signal and its downstream transcription factor GLI1 were upregulated in keloid patient–derived stem-like cells. Inhibition of the HH-GLI1 pathway reduced the expression of genes involved in keloids and fibrosis-inducing cytokines, including osteopontin. Moreover, the HH signal inhibitor vismodegib reduced keloid reconstituted tumor size and keloid-related gene expression in nude mice and the collagen bundle and expression of cytokines characteristic for keloids in ex vivo culture of keloid tissues. These results implicate the HH-GLI1 pathway in keloid pathogenesis and suggest therapeutic targets of keloids.

[1] Department of Molecular Oncology, Institute for Advanced Medical Sciences, Nippon Medical School, Bunkyo-ku, Tokyo 113-8602, Japan. [2] Department of Plastic, Reconstructive and Aesthetic Surgery, Nippon Medical School, Bunkyo-ku, Tokyo 113-8602, Japan. [3]These authors contributed equally: Mamiko Tosa, Yoshinori Abe. ✉email: nobuta@nms.ac.jp

Keloids are a benign dermal fibroproliferative disorder, and their exact etiology is still unclear[1,2]. This disease is particularly prevalent in people of African, Asian, and Hispanic descent, and the familial predominance in its development suggests a genetic predisposition[2]. Moreover, keloids are a disease unique to humans and no animal models for keloids exist.

Keloids are benign inflammatory fibrous tumors that develop in response to skin trauma, and their tissues contain excessive accumulation of extracellular matrix (ECM) components, especially collagen, in the dermis and subcutaneous tissue. These fibroproliferative tumors tend to continue to grow for many years without spontaneous regression, and they often recur with increased disease exacerbation after surgical treatment[3,4]. Because of pain, itching, and limitation of movement around the lesion, keloids significantly impair patient quality of life[5]. Moreover, patients with large or visible keloid scars may face physical, esthetic and psychological consequences, with emotional and financial costs[6]. However, there are many cases of keloids that are refractory or recurrent, and effective treatment for keloids in such cases has not yet been established[7,8].

Keloids are characterized by a raised ectodermal skin outgrowth that extends over the boundaries of the wound through overproduction of collagen type I and III. The histological hallmark of keloids is the presence of thick hyalinized collagen bundles, which is used to distinguish keloids from other abnormal scars[9]. Keloids develop following skin injury and continuous local inflammation, in most cases, months or years after the initial trauma. These processes suggest that skin trauma and inflammation cause dysregulation of the wound healing process, leading to the development of keloids[10,11]. After injury, fibrosis may occur when repair processes are uncontrolled or accentuated, leading to excessive ECM accumulation, which can result in the formation of keloids[10]. Moreover, keloids typically occur at sites where the skin is exposed to external forces, such as the neck, chest, and shoulders, suggesting that the exacerbation of keloid growth is also related to the degree of tension in the tissues surrounding the initial trauma[12]. In addition to these external factors, many studies have shown that keloid fibroblasts differ from normal fibroblasts in the elevated production of ECM, particularly collagen, which is overproduced in keloid scars, and in the response to cytokines and growth factors[13]. Keloid fibroblasts also show alterations in their metabolic system, like cancer cells, in generating ATP mainly from aerobic glycolysis than mitochondrial oxidative phosphorylation[14]. This metabolic reprogramming is one of the hallmarks of cancer for tumor growth[15]. Because these changes are predominantly observed in keloid fibroblasts but not fibroblasts in the same patient's normal tissue, it is possible that epigenetic reprogramming of cellular characteristics occurs in stem cells in an microenvironment that induces keloids, such as inflammation and tissue regeneration. Indeed, altered DNA methylation and histone modifications have been reported in keloid fibroblasts[16].

While keloids are benign skin tumors, they exhibit many cancer-like characteristics, including uncontrolled growth, lack of spontaneous regression, and an extremely high recurrence rate[17,18]. Cancers often arise at sites of chronic injury, and continuous wound healing because of chronic inflammation has been shown experimentally and clinically to be an important factor for cancer development[19]. Moreover, increased fibrosis in tumors promotes tumor cell growth, survival, and malignant transformation, and tumor invasiveness and poor prognosis correlate with the degree of tissue fibrosis and stromal stiffness[20]. Cancers initiate from cancer stem cells (CSCs), which have the ability to form tumors in vivo[21,22]. CSCs are established by impaired differentiation status in tissue stem cells or by stochastic stem cell transformation of differentiated cells caused by various genetic mutations[23]. Like pluripotent stem cells, CSCs are generated and maintained by signals implicated in developmental processes, such as HEDGEHOG (HH), WNT, and NOTCH signaling[24], and reprogramming factors involved in the generation of inducible pluripotent stem cells, such as SOX2, OCT4, KLF4, MYC, and NANOG[25,26]. Furthermore, accumulating evidence has shown that CSC generation by cell reprogramming is induced by chronic inflammation, epithelial–mesenchymal transition (EMT) inducing conditions such as wound healing, and cellular metabolic changes[27–29]. During oncogenesis, CSCs are generated by cellular reprogramming through epigenetic changes under these conditions, and CSCs are thought to exhibit plasticity, changing from stem-like cells to non-tumorigenic differentiated cancer cells or vice versa[30]. Studies have clarified that keloids do not turn into cancers[31], but it is possible that conditions in keloid tissues could epigenetically alter the stem cells for skin tissue[32]. In this context, pathological stem cells were thought to be present in keloid tissue[33], and a previous study showed that mesenchymal-like stem cells surrounded by an inflammatory niche in keloid tissues sustained keloid growth[34,35]. However, the detailed characteristics of keloid stem cells and their role in disease remain to be elucidated.

The HH signal is one of the fundamental signaling pathways that contributes to epidermal development, homeostasis, and repair and to epidermal stem cell maintenance[36]. Moreover, dermal cells are considered key determinants in tissue regeneration during wound healing, and activation of the dermal HH signal is important for neogenesis of dermal papilla[37]. In addition to the regenerative activity of the HH signal in skin wound healing, high expression of glioma-associated oncogene 1 (GLI1)[38], the effector and transcriptional activator of HH signaling[39], was observed, suggesting the role of HH-GLI1 signaling pathway in keloids. The GLI family transcription factors include three members, GLI1, GLI2, and GLI3. Studies in gene knockout mice revealed that GLI2 and GLI3 are the major activator and repressor of HH signaling, respectively, whereas GLI1 likely serves as a signal amplifier of GLI2[39,40]. HH-GLI signaling is an important pathway involved in development and cancer, and total activity of GLIs regulates the fate and behavior of stem cells and cancer stem cells[41]. GLI1 is expressed at low levels in differentiated tissues, and aberrant activation of GLI1 plays a role in the promotion of cancer through regulation of cell proliferation, survival, metastasis, and generation of cancer stem cells[42].

In this study, we found that the HH signaling pathway and its downstream effector, the transcription factor GLI1, are activated in keloid stem-like cells derived from patients and suppression of HH signaling inhibits keloid-like tissue progression in a mouse transplantation assay. Therefore, our results provide insights into the molecular mechanism of keloid pathogenesis and the possibility of keloid treatment by inhibition of the HH-GLI1 signal.

## Results

**The HH signaling pathway is upregulated in keloid fibroblast–derived stem-like cells**. Different keloid regions exhibit different growth characteristics; the central portion of the keloid is a red, elastic, hard, raised lesion, (termed the "fibrosis parts of keloid"), while the marginal (peripheral) regions tend to grow and invade into normal skin, termed the "proliferative" or "invasive" keloid regions[43]. To examine the characteristics of stem-like cells from keloid tissues compared with normal skin, we obtained biopsy samples from tissues in the keloid marginal area, keloid central area, and normal-looking skin adjacent to the lesion (Fig. 1a). To identify the gene expression signatures of fibroblasts and stem-like cells from keloid legions compared with nonaffected donors, we performed 3D principal component

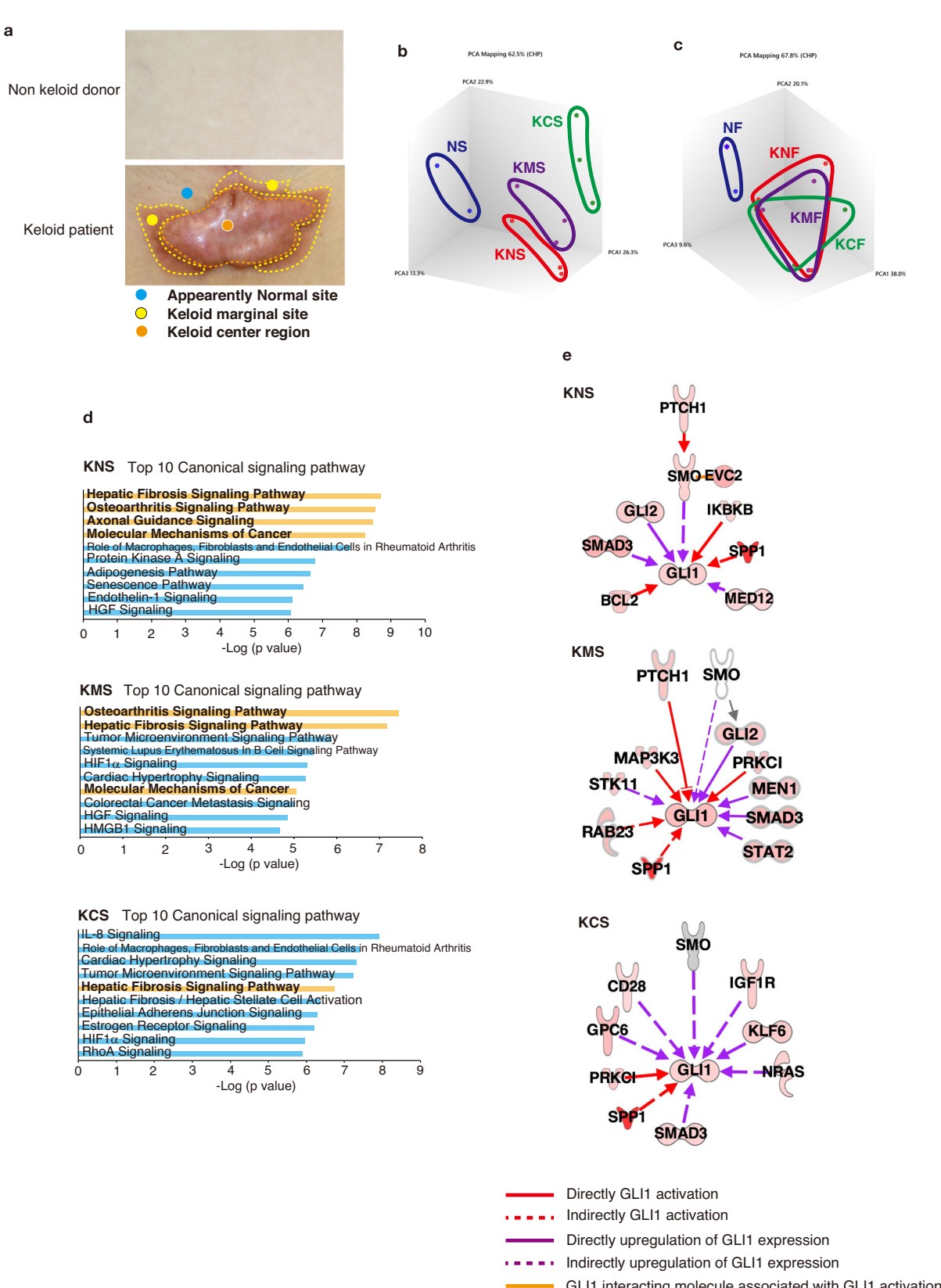

analysis (PCA) of the transcriptome from keloid fibroblasts in adherent culture and stem-like cells that grow in neural sphere-forming conditions[35]. We examined the expression of reprogramming factors OCT4 and NANOG which regulate a cascade of pathways to control pluripotency, self-renewal, genome surveillance, and cell fate determination[44]. As shown in Supplementary Fig. 1a–d, we found that expression of OCT4 and NANOG proteins was not detected in fibroblasts in and around the keloid sites, and in normal fibroblasts, but most of the sphere-forming cells in these sites expressed these proteins at high levels. In contrast, the expression of OCT4 and NANOG proteins was increased in sphere-forming cells, while mRNA expression was

**Fig. 1 The HH signaling pathway is activated in keloid fibroblast-derived stem-like cells. a** Biopsy site from normal skin (non-wounded), keloid center, keloid marginal area, and normal skin adjacent to the keloid from which in vitro primary fibroblasts (transcriptome profiling of normal fibroblasts [NF], fibroblasts from normal-looking skin adjacent keloids [KNF], keloid marginal area [KMF], and keloid central area [KCF], and fibroblast-derived stem-like cells (normal dermis [NS], keloid normal dermis [KNS], keloid marginal area [KMS], and keloid central area [KCS]) were subsequently established. **b** 3D principal component analysis (PCA) from keloid fibroblast-derived stem cell transcriptome revealed distinct changes across three keloid tissue areas (KNS, KMS, KCS). Transcriptome data was obtained from microarray assay using two donors (N1, N2) with normal dermis and three keloid patients (K1, K2, K3). **c** 3D PCA from keloid fibroblast and normal fibroblast transcriptome revealed that keloid fibroblasts exhibit similar transcriptome profiles across three areas (KNF, KMF, KCF) in keloid tissue. Transcriptome data was obtained as described in (**b**). **d** Ingenuity pathway analysis (IPA) showing the top 10 canonical signaling pathways associated with pathogenesis in genes upregulated in KNS, KMS, or KCS vs. NS. Each bar represents $p$ values. The HH signaling pathway components were upregulated in canonical signaling pathways shown in the bold orange bar. **e** IPA revealed that KNS, KMS, and KCS have different upregulated genes associated with GLI1 activation. Upregulated genes associated with GLI1 activation in KNS, KMS, or KCS are shown in red. Unchanged genes are shown in gray. The role of the relationship is shown in the figure.

induced but not as significant as these proteins (Supplementary Fig. 1e), which may be due to the finding that the expression of these reprogramming factors in stem cells is mainly regulated by post-transcriptional modifications[45]. Moreover, we analyzed gene expression profile in sphere-forming cells from microarray analysis and found upregulation of the signaling pathways for Human Embryonic Stem Cells Pluripotency by Ingenuity Pathway Analysis (IPA) software (Supplementary Fig. 1f). These results suggest that the sphere-forming cells are homogeneous and have stem cell-like characteristics. However, we did not find any stem cell markers previously reported in several experimental systems[35,46] that are specifically increased in sphere-forming cells (Supplementary Fig. 1g–m). These results did not conclude that the sphere-forming cells were stem cells but suggested that they were cells with a stem cell-like phenotype. As shown in Fig. 1b, c, transcriptome profiling of normal fibroblasts (hereinafter referred to as NF) and normal stem-like cells (NS) in dermis from non-affected donors was different from that of keloid fibroblasts (KF) and stem-like cells in keloid tissues. In contrast, transcriptome profiling was not significantly different in fibroblasts from normal-looking skin adjacent keloids (KNF), keloid marginal area (KMF), and keloid central area (KCF), but it showed distinct changes between stem-like cells from keloid normal dermis (KNS), keloid marginal area (KMS), and keloid central area (KCS) (Fig. 1b). The normal skin adjacent to keloid has shown to be partially characterized by keloid, as its fibroblasts produce high levels of TGF-β and collagen[9].

To investigate the signaling pathways that may cause keloid pathogenesis, the canonical pathways activated in these cells were assessed by IPA software. The top 10 pathways identified in keloid patient–derived stem-like cells are illustrated in Fig. 1d. Among these pathways, pathways involved in the HH signal (see detail in Supplementary Fig. 2) were activated in KNS (4 of the top 10 canonical signaling pathways in KNS) compared with NS (Fig. 1d). Notably, these pathways include signaling pathways of hepatic fibrosis and osteoarthritis that lead to the accumulation of ECM proteins and fibrosis characteristic of keloids. These two signaling pathways were also most active in KMS, while they were not so active in KCS (Fig. 1d). Upstream regulator analysis revealed that the expression of several molecules involved in GLI1 activation was elevated in these three types of stem-like cells in keloid patients, although not completely identical (Fig. 1e).

**GLI1 is upregulated in stem-like cells in keloid tissues.** To explore the role of GLIs in keloid pathogenesis, we next analyzed the expression of GLIs in fibroblasts and stem-like cells from keloid patients and found that the expression of GLI1, but not GLI2 and GLI3, was enhanced specifically in stem-like cells from keloid tissues and also adjacent normal-looking tissues, but not in keloid fibroblasts (Fig. 2a, b). The GLI1-expressing cells were present in similar proportions in keloid-derived fibroblasts, and

their numbers were elevated by SAG, a SMO agonist that activates the HH pathway[47] (Supplementary Fig. 3a–c). Moreover, the protein expression of pluripotency factors to drive stemness, OCT4 (encoded by *POU5F1*) and NANOG were upregulated in KNS and KMS (Supplementary Fig. 4). In tissues, relatively higher amounts of cells expressing the stem cell marker OCT4 or NANOG were observed in keloid tissues compared with normal tissues (Fig. 2d–g and Supplementary Fig. 4a, b), and GLI1-expressing cells were present in similar proportions as cells expressing both GLI1 and OCT4 or NANOG[48] (Fig. 2e, g, compared with Supplementary Fig. 4a, b). Furthermore, the increased expression of GLI1 mRNA observed in sphere-forming cells was not observed in fibroblasts cultured under the same conditions (Supplementary Fig. 3d), suggesting that the activation of GLI1 in sphere-forming cells was not due to differences in cell culture conditions, but rather to their change into stem-like cells. These results suggest that the expression of GLI1 plays a role in stem-like cells in tissues from keloid patients.

**SHH is widely expressed in keloid and surrounding dermal tissues.** Three HH ligands have been identified in mammals: sonic hedgehog (SHH), Indian hedgehog (IHH), and desert hedgehog (DHH). SHH plays a role in skin development and regeneration, maintaining stem-like cells, and regulating the development of hair follicles and sebaceous glands[36]. Because GLI1 is a target gene of the HH signal pathway[39], we further analyzed the expression of SHH in keloid tissue and found enhanced protein expression of SHH in keloid tissue and surrounding normal-looking tissues compared with normal skin and mature scar from a non-keloid patient (Fig. 3a–d) and by immunocytochemistry, SHH was mainly expressed in the cytoplasm (Supplementary Fig. 5). Furthermore, expression of the main HH target gene *PTCH1* was enhanced in stem-like cells (KNS, KMS and KCS) compared with keloid fibroblasts (Fig. 3e, f). The expression of *PTCH1* was enhanced in GLI1-upregulated keloid-derived stem-like cells (see Fig. 2a) but not in fibroblasts from keloid tissues (Fig. 3e, f), suggesting that the response to SHH differs between mature fibroblasts and stem-like cells. Moreover, the expressions of *PTCH1* and *GLI1* were downregulated in the flat quiescent center area without erythema[9] (Fig. 3g, h), suggesting that the keloid lesions improved with decreased HH signaling. In relation to this, the mRNA expression of a key HH signal transducer smoothened (SMO) was enhanced in stem-like cells (Fig. 3i). However, the mechanism by which SMO mRNA expression is higher in NS than in stem-like cells of keloid patients has not been elucidated.

**The HH-GLI1 signaling and fibrosis-related pathways are activated in keloid tissues and keloid fibroblast-derived stem-like cells.** We examined upregulated GLI1 target genes in keloid stem-like cells by IPA, and 16 genes were identified as putative

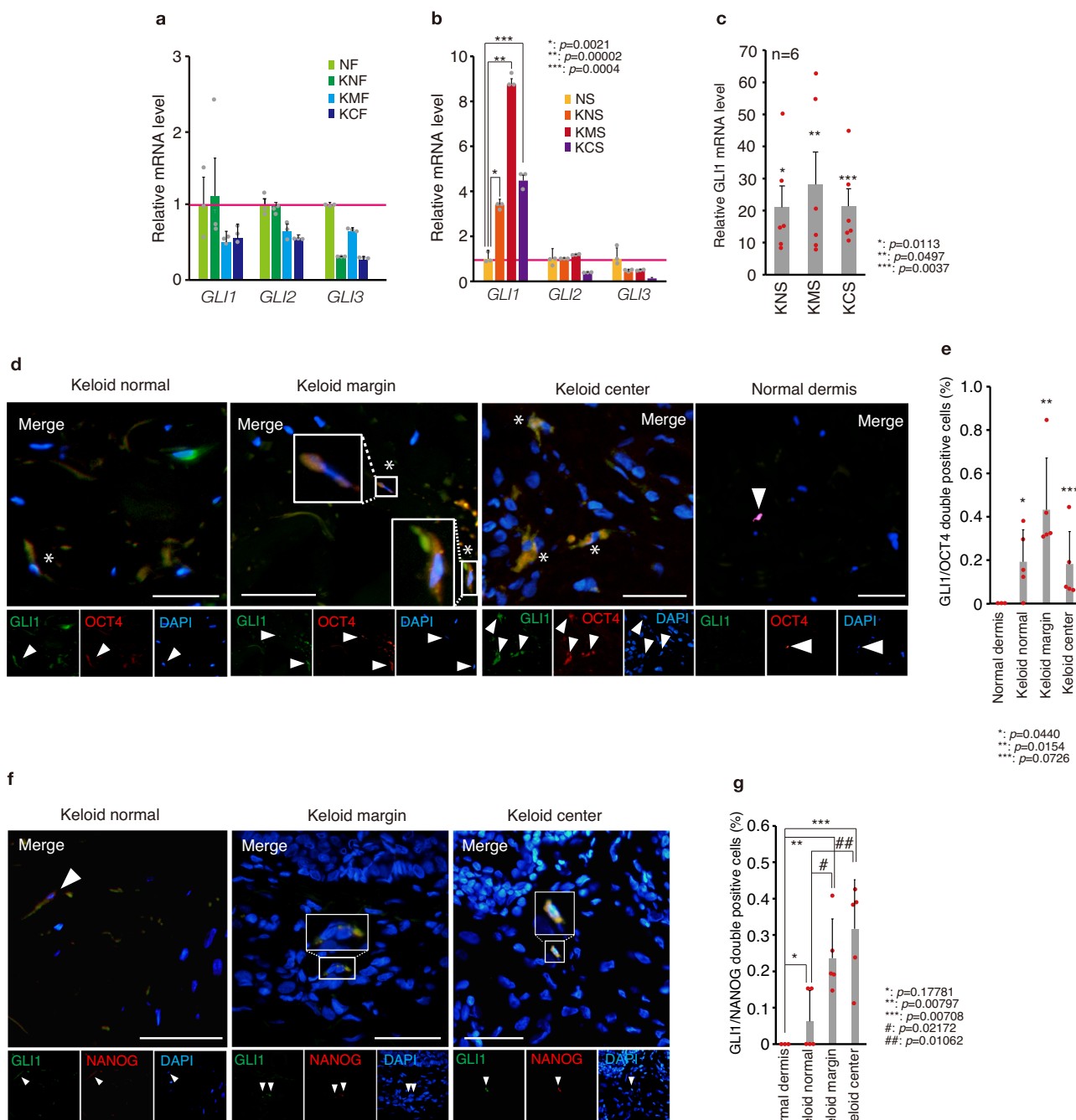

**Fig. 2 GLI1 expression is upregulated in keloid fibroblast-derived stem-like cells. a** qPCR analysis revealed that the mRNA expression of GLI family members (GLI1, GLI2, GLI3) was reduced in keloid fibroblasts compared with that in normal fibroblasts. **b** qPCR analysis revealed that GLI1 mRNA expression was exclusively upregulated in keloid fibroblast-derived stem-like cells compared with that in normal fibroblast-derived stem-like cells. In (**a**) and (**b**), total RNA was obtained from K1 (passage 2) and N1 (passage 2)-derived stem-like cells. Results are shown as means ± S.D. from triplicated experiments. **c** qPCR analysis revealed that GLI1 mRNA expression is expressed at the highest levels in keloid fibroblast-derived stem-like cells. Results are shown as fold change of GLI1 mRNA expression (patient K1, K2, K4, K5, K6, and K8 [n = 6], passage 2) in keloid fibroblasts compared with GLI1 mRNA expression in normal fibroblasts (N1, passage 2). Data are shown as mean ± S.D. from triplicated experiments. **d**, **f** Representative immunofluorescent staining of GLI1 and OCT4 co-expression (**c**) or GLI1 and NANOG co-expression (**e**) in keloid tissue. Green indicates GLI1; red indicates OCT4 or NANOG. DNA was stained with DAPI (blue). The asterisk shows GLI1 and OCT4 or GLI1 and NANOG double-positive cells. Arrowhead shows GLI1-, OCT4-, or NANOG-positive cells. Bar, 500 μm. **e**, **g** Quantification of GLI1 and OCT4 (**d**) or NANOG (**f**) co-expressing cells in three areas of keloid tissue and normal dermis (keloid tissue: K8, K9, K10, K11, K12 [n = 5]; normal dermis: N3, N4, N5 [n = 3]). Cells were counted from five independent views (one view: 0.35 mm × 0.26 mm). Results are shown as means ± S.E.M. from five patient-derived keloid tissue and three donor-derived normal dermis, and the dot indicates the result from each patient or donor. Patient information is shown in Supplementary Table 1; source data are provided as a Source Data file 1.

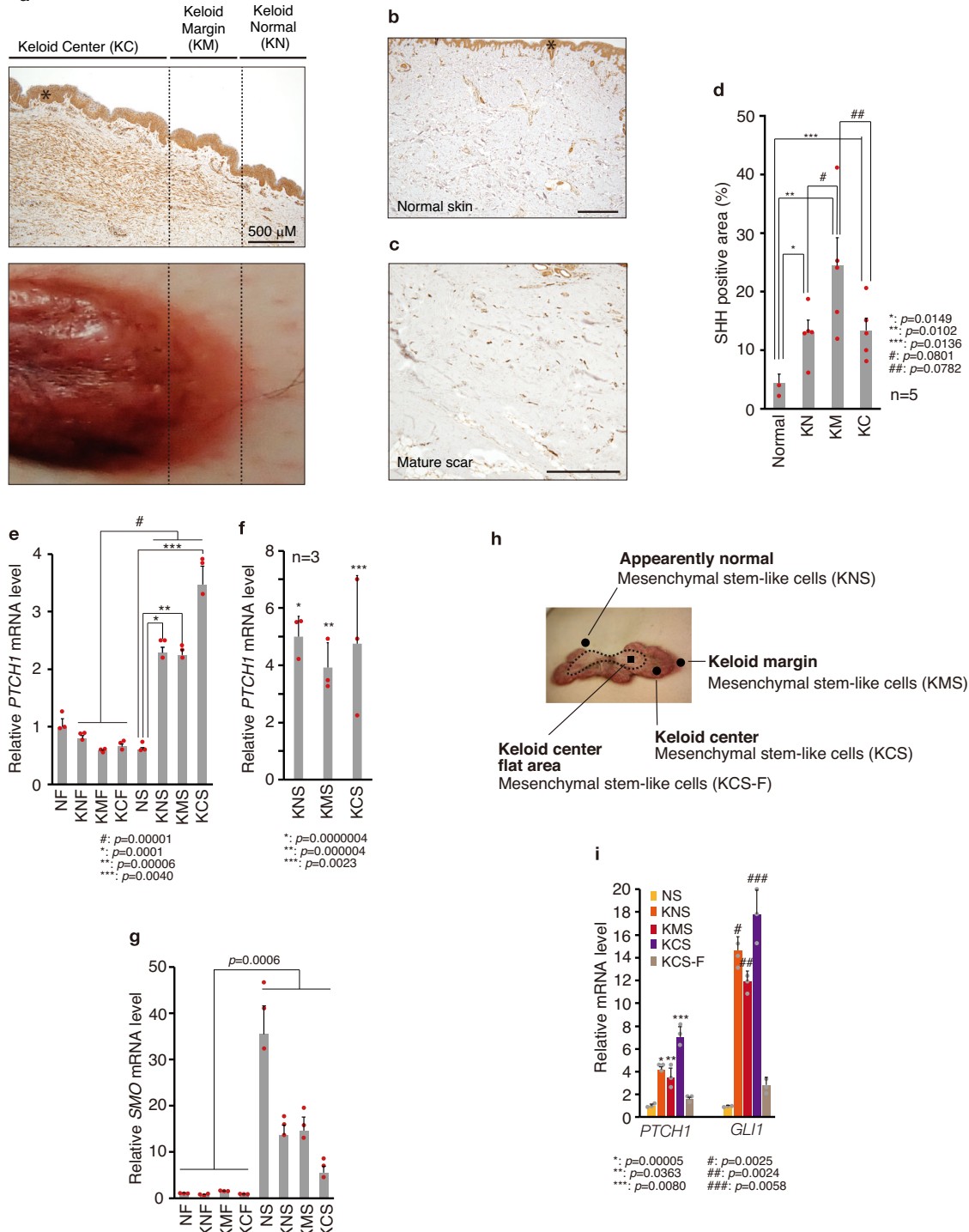

**Fig. 3 SHH is widely expressed in keloid tissue, and PTCH1 expression is expressed at the highest levels in keloid fibroblast-derived stem-like cells.**
Representative immunohistochemical staining of SHH expression in keloid tissue (**a**: K10), normal dermis (**b**: N3), and mature scar (**c**: MS1). Bar, 500 μm. The lower image is a representative image of keloid from the surface view. The asterisk represents a non-specific signal. The magnified image is shown in Supplementary Fig. 3a. **d** Quantification of the SHH-positive area in three areas of keloid tissue, normal dermis, and mature scar (keloid tissue K1, K2, K3, K4, K6, K9 [n = 6], normal dermis N3, and mature scar MS1). The positive area was quantified from a view (0.5 mm × 0.5 mm) in each keloid area or normal/mature scar dermis. Results of the SHH-positive area in the normal dermis and mature scar are combined and shown as "Normal." Data are shown as means ± S.E.M. **e**, **f** qPCR analysis reveals that PTCH1 mRNA expression is the most abundant in keloid fibroblast-derived stem-like cells (K7, passage 2). In (**f**), qPCR analysis shows fold change of PTCH1 mRNA expression in keloid fibroblast-derived stem-like cells (K3, K5, K12 [n = 3]) compared with normal fibroblast-derived stem-like cells (N1). **g** Representative keloid center flat area viewed from surface skin. **h** qPCR analysis revealed that PTCH1 expression is downregulated in stem-like cells derived from a flat area in the keloid center. **i** qPCR analysis showed that SMO is widely expressed in stem-like cells (K7, passage 2). In (**e**, **f**, **h**, **i**), results are shown as means ± S.D. from triplicated experiments. Patient information is shown in Supplementary Table 1; source data are provided as a Source Data file 1.

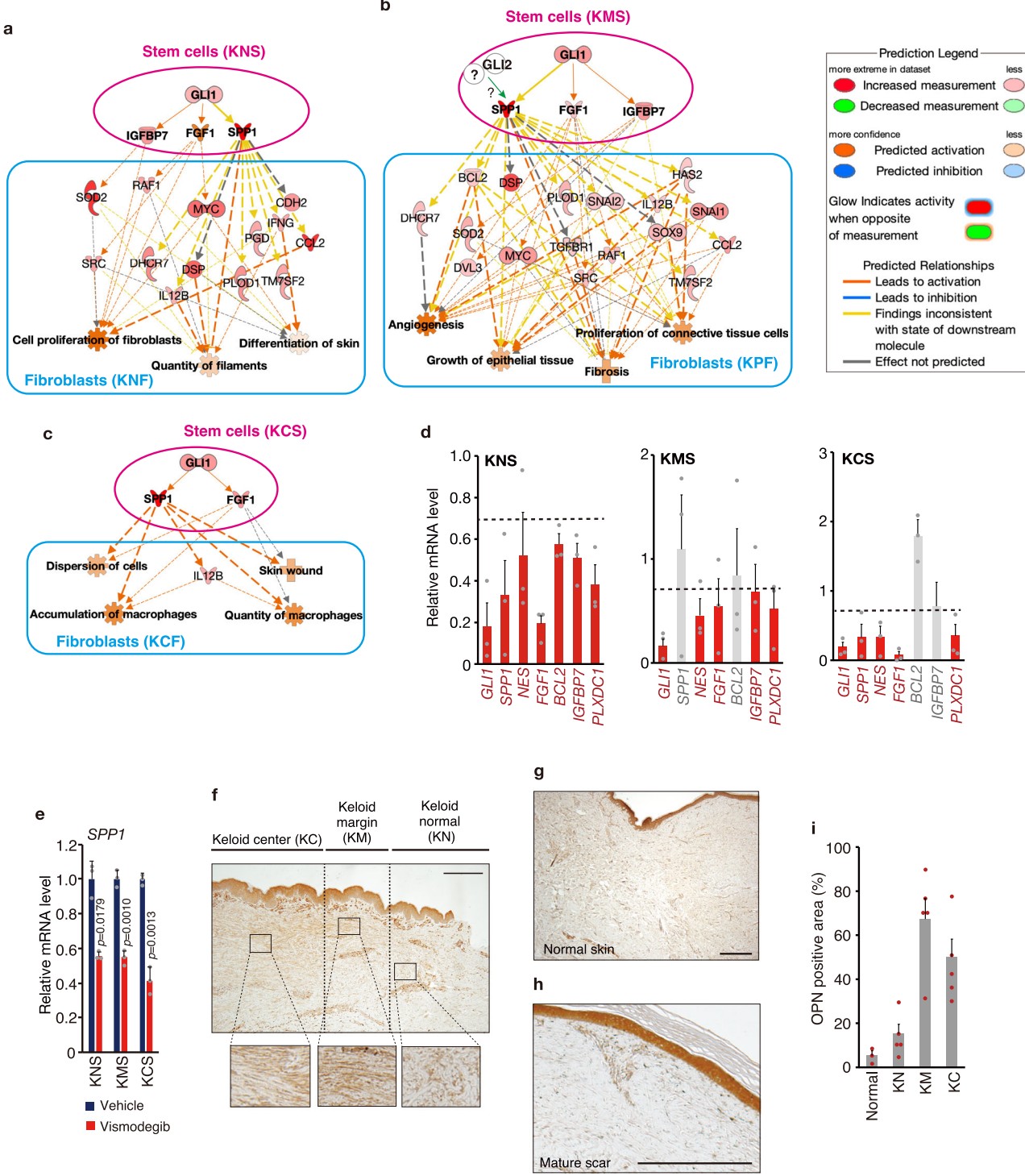

GLI1 target genes (Supplementary Fig. 6a). We then validated that 7 of the 16 genes were GLI1 target genes in keloid stem-like cells from results showing that mRNA expression of the 7 genes was upregulated in keloid stem-like cells (Supplementary Fig. 6b). We analyzed gene expression profiling downstream of HH signaling in keloids. IPA revealed upregulation of GLI1 target secretory cytokines involved in tissue fibrosis, osteopontin (OPN, encoded by *SPP1, secreted phosphoprotein 1*)[49], fibroblast growth factor 1 (FGF1)[50], and insulin-like growth factor binding protein-7 (IGFBP7)[51], in keloid-derived stem-like cells; these cytokines regulated signals in keloid-derived fibroblasts (Fig. 4a–c and Supplementary Fig. 7a–c). The expressions of major target gene

*SPP1* (encoding OPN) (Supplementary Fig. 6b) and other targets in keloid stem-like cells were confirmed by mRNA analysis (Fig. 4d). Most of the mRNA expression of these cytokines and other targets in KNS, KMS and KCS were suppressed by GLI1 knockdown (Supplementary Fig. 7d–f). Moreover, the expression of *SPP1* (encoding OPN) and other targets was suppressed by vismodegib[52], an inhibitor of the HH pathway that binds to SMO (Fig. 4e and Supplementary Fig. 7g). We also confirmed that the expression of OPN protein, which is an HH/GLI1-inducible protein[53], was upregulated in keloid tissues, especially in the keloid marginal area, but not in normal skin and mature scar from non-keloid donor (Fig. 4f, g). These findings suggest the

**Fig. 4 Site-specific GLI1 and its target genes function in stem-like cells and fibroblasts. a–c** IPA suggested that GLI1 target secretory molecules (SPP1 and FGF1) from stem-like cells provide site-specific keloid fibroblast function (KNS–KNF [keloid normal] axis: **a** KMS–KMF [keloid marginal area] axis: **b** KCS–KCF [keloid central area] axis. **c** Downstream GLI1 target genes were selected from microarray analysis shown in Supplementary Fig. 4. Upregulated genes in keloid fibroblast-derived stem-like cells were compared with those in normal fibroblast-derived stem-like cells. The state of gene expression changes and the relationship are shown in the figure. The solid line represents a direct relationship, and the dashed line represents an indirect relationship. Bold solid/dashed lines represent OPN-mediated regulation of gene expression. **d** Average suppression level of GLI1 target gene expression in keloid fibroblast-derived stem-like cells from three patients (K11, K14, K9 [$n = 3$]). Data are shown as mean ± S.E.M. from results shown in Supplementary Fig. 4a–c. GLI expression was suppressed more than 30% by knock down, as shown in red. **e** qPCR analysis revealed that treatment with a SMO inhibitor, vismodegib, suppressed *SPP1* (OPN) gene expression in keloid fibroblast-derived stem-like cells. Cells (10,000, patient K8, passage 2) were seeded in 6-well ultra-low attachment plates and incubated for 4 days. Next, 10 μM vismodegib was treated for 4 days. Results are shown as means ± S.D. from triplicated experiments. The result of other GLI1 target gene expression after vismodegib treatment is shown in Supplementary Fig. 5g. Representative image of OPN expression in keloid tissue (**f**), normal dermis (**g**), and mature scar (**h**). Bar, 500 μm. OPN is shown in brown. The asterisk represents the non-specific signal. **i** OPN expression was expressed at the highest levels in keloid marginal and center areas. The result shows the positive area in three areas of keloid tissue, normal dermis, and mature scar (keloid tissue: K1, K2, K3, K4, K9 [$n = 5$], normal dermis: N1 [$n = 1$], mature scar: MS1, MS2 [$n = 2$]). The positive area was quantified from a view (0.5 mm × 0.5 mm) in each keloid area or normal/mature scar dermis. Results of OPN-positive area in the normal dermis and mature scar were combined and shown as "Normal." The result is shown as means ± S.E.M. Patient information is shown in Supplementary Table 1; source data are provided as a Source Data file 1.

possibility that HH-GLI1 pathway regulates keloid fibrotic characteristics through OPN and other cytokines.

**Inhibition of the HH-GLI1 pathway reduces the number of keloid fibroblast-derived stem-like cells**. The HH-GLI signaling pathway regulates stem cell maintenance and cell fate determination during development, tissue regeneration, and ongogenesis[39,41]. Thus, we next analyzed whether the enhanced HH-GLI1 pathway is involved in the regulation of stem-like cells in keloid tissues. As shown in Fig. 5a, b and Supplementary Fig. 8a, GLI1 knockdown reduced the number of sphere-forming cells, a standard method for determining the stem cell' population[35]. Knockdown of SMO also inhibited stem cell number (Fig. 5c, d and Supplementary Fig. 8b), suggesting that the HH-GLI1 pathway maintains stemness of keloid stem-like cells. Next, to analyze the effect of HH-signal inhibition in the viable cell number of stem-like cells, sphere forming cells were treated by HH-GLI1 pathway inhibitors (Fig. 5e). The number of living stem-like cells was marginally reduced by vismodegib (Fig. 5f and Supplementary Fig. 8c) or GLI1/2 inhibitor (GANT61; Fig. 5g). Furthermore, HH pathway inhibition did not affect the viability of normal fibroblast-derived sphere forming cells (Supplementary Fig. 8d). Since GL1 or SMO knockdown resulted in a relatively high degree of suppression of the number of sphere-forming cells, whereas the suppression of the number of viable fibroblasts by HH pathway suppression was weak (Supplementary Fig. 8e–h), these results suggest that suppression of HH-GLI1 signaling is related to stem cell maintenance rather than to the survival of stem-like cells.

**HH signaling pathway is involved in disease progression of keloid model**. The lack of appropriate animal models has made it difficult to examine the efficacy of treatments on keloid tissue. A recent study reconstructed human keloid-like tissue by transplanting fibroblasts from keloid tissue into nude mice[54]. We used this mouse transplant model to analyze the effect of HH signaling inhibitors on keloid tissue formation (Fig. 6a). The volume of reconstructed keloid-like tissue in nude mice was suppressed by vismodegib (Fig. 6b). On the other hand, while statistically significant, these in vivo results have considerable variability, making it difficult to estimate the magnitude of the effect of vismodegib. Therefore, we used various analyses and experiments to examine the effects of vismodegib on keloids. First, the expressions of genes highly upregulated in keloid tissue, *IL-6*, *CTGF* and *COL1A2*, were enhanced in keloid marginal fibroblast–transplanted tissue, and this expression was suppressed

by vismodegib (Fig. 6c). Moreover, gene set enrichment analysis (GSEA) revealed that the expression of genes associated with keloid pathogenesis (growth factor binding, extracellular structure organization, bone morphogenesis, collagen fibril organization) was suppressed by vismodegib (Fig. 6d). Furthermore, production of hyalinized collagen bundles, characteristic of keloid fibrosis tissue[55], was reduced by vismodegib (Supplementary Fig. 9). These results suggest the involvement of HH signal in keloid pathogenesis.

**Characteristics involved in the pathogenesis of keloids are suppressed by vismodegib in keloid organ culture**. To further confirm the effect of vismodegib in human keloid tissues, we used a keloid ex vivo organ culture system[56]. Keloid tissue was embedded in collagen gel layered with culture medium with or without vismodegib (Supplementary Fig. 10a). HH signaling is known to crosstalk with TGF-β and WNT signaling pathways that are involved in fibrosis[57]. Therefore, we investigated whether the enhanced HH signaling observed in keloid tissues affects well-known pro-fibrotic molecules. However, as shown in Supplementary Fig. 10d–f, the expression of the WNT-signaling molecule β-CATENIN, TGF-β and its effector transcription factor SMAD3 in keloid tissue was not significantly altered among four patients-derived tissue. Next, we analyzed the effect of vismodegib in this system, hyalinized collagen bundles were clearly decreased by vismodegib (Fig. 7a, b and Supplementary Fig. 11a, b). In addition, we also found that the expression of TGF-β, but not β-CATENIN, was suppressed (Supplementary Fig. 12a, b), suggesting that TGF-β signaling is activated downstream of HH signaling. Moreover, the expression of proteins involved in keloid pathogenesis and disease state, IL-6 and connective tissue growth factor (CTGF)[58], and OPN were also suppressed by vismodegib (Fig. 7c–h). These results suggest the possibility that inhibition of the HH-GLI1 signal may be effective in treatment against keloid progression and recurrence.

## Discussion

In the present study, we found that the characteristics of stem-like cells from keloid lesions and surrounding dermis differed from those of normal skin, with increased activity of HH signaling and its downstream transcription factor GLI1. Furthermore, we found that SHH protein is widely expressed in keloids and surrounding dermal tissues. Furthermore, we found that inhibition of the HH-GLI1 pathway decreased the number of keloid stem-like cells in cultured keloid fibroblasts and in vivo reconstructed keloid tissue in nude mice, and decreased the expression of genes involved in keloids and fibrosis-inducing cytokines that induce

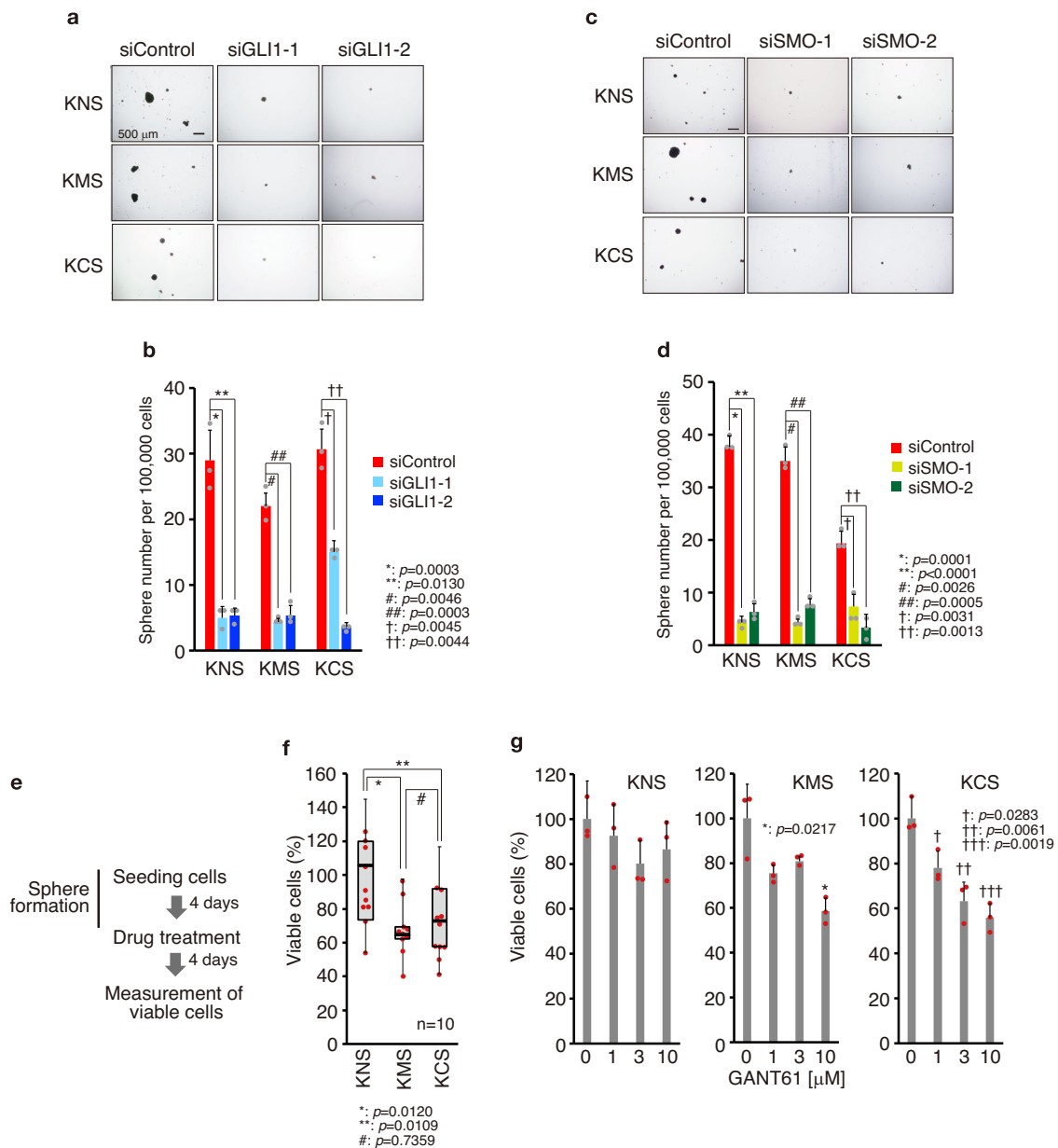

**Fig. 5 Inhibition of the HH signaling pathway reduces viable keloid fibroblast-derived stem-like cells. a** Representative image of control and GLI1-knocked down spheres (keloid fibroblast-derived stem like cells). siRNAs for GLI1 (siGLI1-1 and siGLI1-2) were stably expressed by recombinant lentivirus. GLI1-knocked-down cells were obtained by puromycin selection. Bar, 500 μm. **b** Quantification of spheres from (**a**). Fibroblasts (100,000, patient K14, passage 2) were seeded in 6-well ultra-low attachment plates for sphere formation in triplicate. Ten days later, numbers of spheres (~ >200 μm sphere diameter) were counted. **c** Representative image of control and SMO-knocked down spheres. SMO siRNAs (siSMO-1 and siSMO-2) were stably expressed by recombinant lentivirus. SMO-knocked down cells were obtained by puromycin selection. Bar, 500 μm. **d** Quantification of spheres from (**c**). Cells (100,000, patient K14, passage 2) were seeded in 6-well ultra-low attachment plates. The number of spheres was counted as described in (**b**). In (**b**) and (**d**), results are shown as means ± S.D. from triplicated experiments. **e** Schematic illustration of the experimental design in (**f**) and (**g**). **f** Quantification of viable stem-like cells after SMO inhibitor treatment. Fibroblasts (5000, passage 1) were seeded in 96-well-flat bottom ultra-low attachment plates for stem cell culture. Four days later, 10 μM vismodegib (SMO inhibitor) was treated for 4 days. Quantification of viable stem-like cells was performed using the Cell Counting Kit-8. The box plot shows viable cell maximum, minimum, and median percentages. Bold lines on each box plot define the median value (K8, K9, K10, K11, K13, K14, K17, K18, K19, K20 [n = 10]). **g** Quantification of viable stem-like cells after GLI inhibitor treatment. Cells (5000, K15, passage 1) were seeded in 96-well-round bottom ultra-low attachment plates for stem cell culture. Four days later, 10 μM GANT-61 (GLI1/2 inhibitor) was treated for 4 days. Quantification of viable stem-like cells was performed as described in (**f**). Patient information is shown in Supplementary Table 1; target sequences for GLI1 or SMO are shown in Supplementary Table 2; source data are provided as a Source Data file 1.

collagen expression. Simultaneously, the HH signaling inhibitor vismodegib reduced reconstructed keloid tumor size and keloid-related gene expression in nude mice, and reduced collagen bundles and expression of cytokines characteristic of keloids in ex vivo cultures of keloid tissue. Although we did not provide a

mechanism as to why SHH expression is increased in keloid tissues and by what mechanism activation of SHH signaling alters properties of stem like cells, our results suggest that the HH-GLI1 pathway is involved in the pathogenesis of keloids. The signaling properties of stem-like cells in keloid lesions and the surrounding

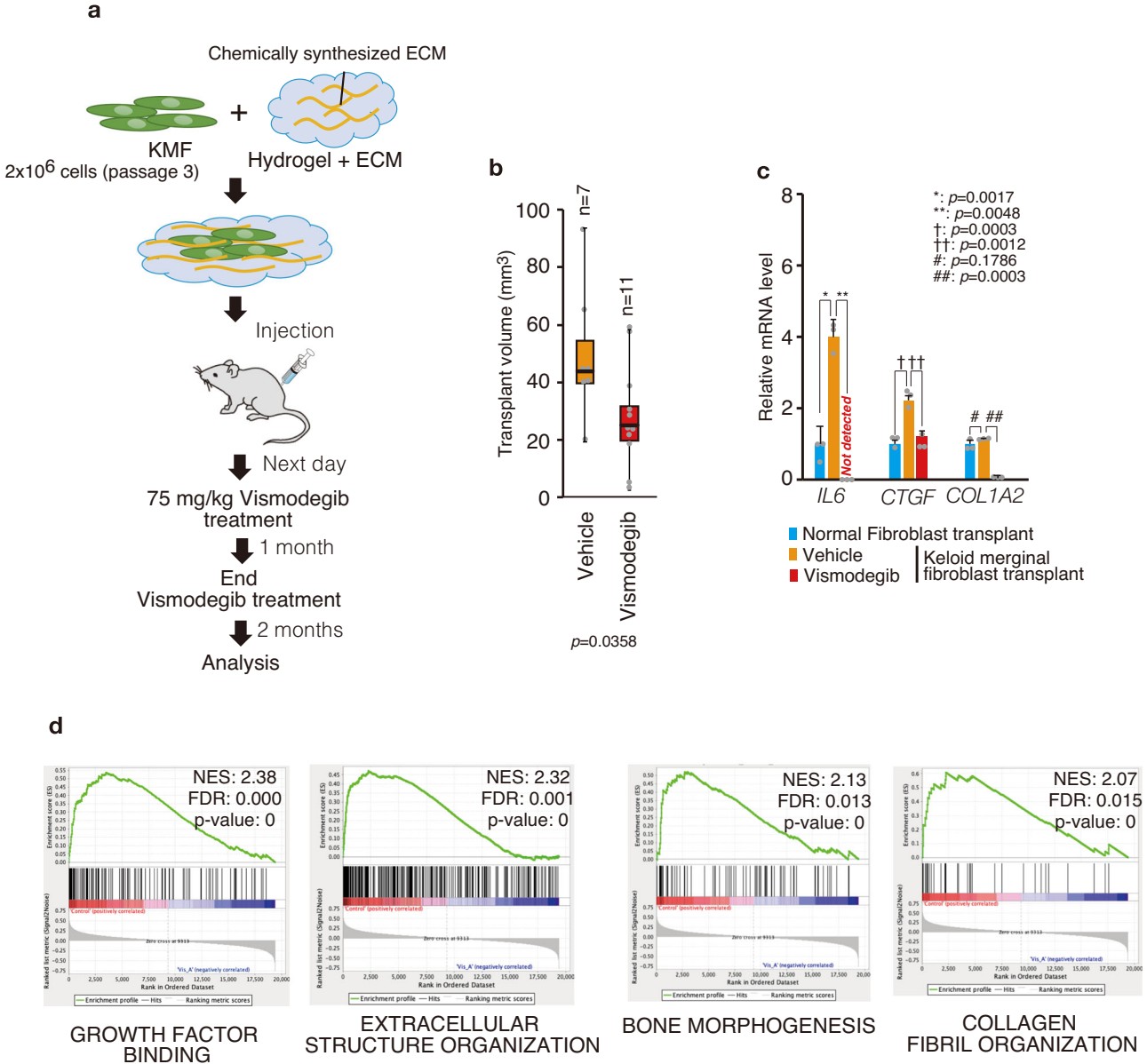

**Fig. 6 The HH signaling pathway is involved in maintaining the keloid tissue. a** Schematic illustration of the experimental design for keloid marginal region-derived fibroblast implantation and drug treatment. **b** Vismodegib treatment reduced xenografted transplant volume. Each box plot shows the maximum, minimum, and median transplant volume. Bold lines on each box plot define the median value from seven transplants from two donor-derived normal fibroblasts (N3, N4) and 11 randomly selected transplants from three patient (K12, K13, K18)-derived keloid marginal fibroblasts. The dot represents the transplant volume from each patient-derived keloid marginal fibroblast xenograft. **c** qPCR analysis reveals that the expressions of keloid-associated genes (IL-6, CTGF, and COL1A2) were downregulated by SMO inhibitor treatment in keloid marginal fibroblast (patient: K21)-xenografted transplant. Results are shown as means ± S.D. from three patient-derived transplants and one donor-derived transplant. **d** Gene set enrichment analysis (GSEA) comparing vehicle treatment and SMO inhibitor treatment for enrichment (growth factor binding, extracellular structure organization, bone morphogenesis, collagen bundle organization) associated with keloid pathogenesis. Transcription profiles were obtained by microarray analysis from keloid marginal fibroblast (K12, K13, K18 [n = 3])-xenografted transplant and normal fibroblast (N3, N4 [n = 2])-xenografted transplant. Patient information is shown in Supplementary Table 1; source data are provided as a Source Data file 1.

cells including fibroblasts revealed in this study are summarized in Fig. 8. We found that the secretion of SHH and expression of GLI1 and its target genes are high at the keloid margins, where cell proliferation is very active. Indeed, SHH has been shown to play an important role in skin wound healing and hair follicle regeneration[59,60]. Moreover, other studies found that long non-coding RNAs suppressing HH signaling and GLI1 are upregulated in keloids[38,61]. Because keloids are thought to develop in a dysregulated wound healing process[62], it is possible that

overproduced SHH, which acts as a morphogen within developing tissue[63], induces keloids through modulation of stem-like cells in a process distinct from normal regulation. SHH was shown to regulate inflammation and tissue regeneration occurring in skin wound repair[64], suggesting that SHH production is induced during the wound healing process that causes keloids. Genetic predisposition is involved in the development of keloids[1,2]. It is also possible that SHH production during wound healing and susceptibility to it is differentially regulated by genetic

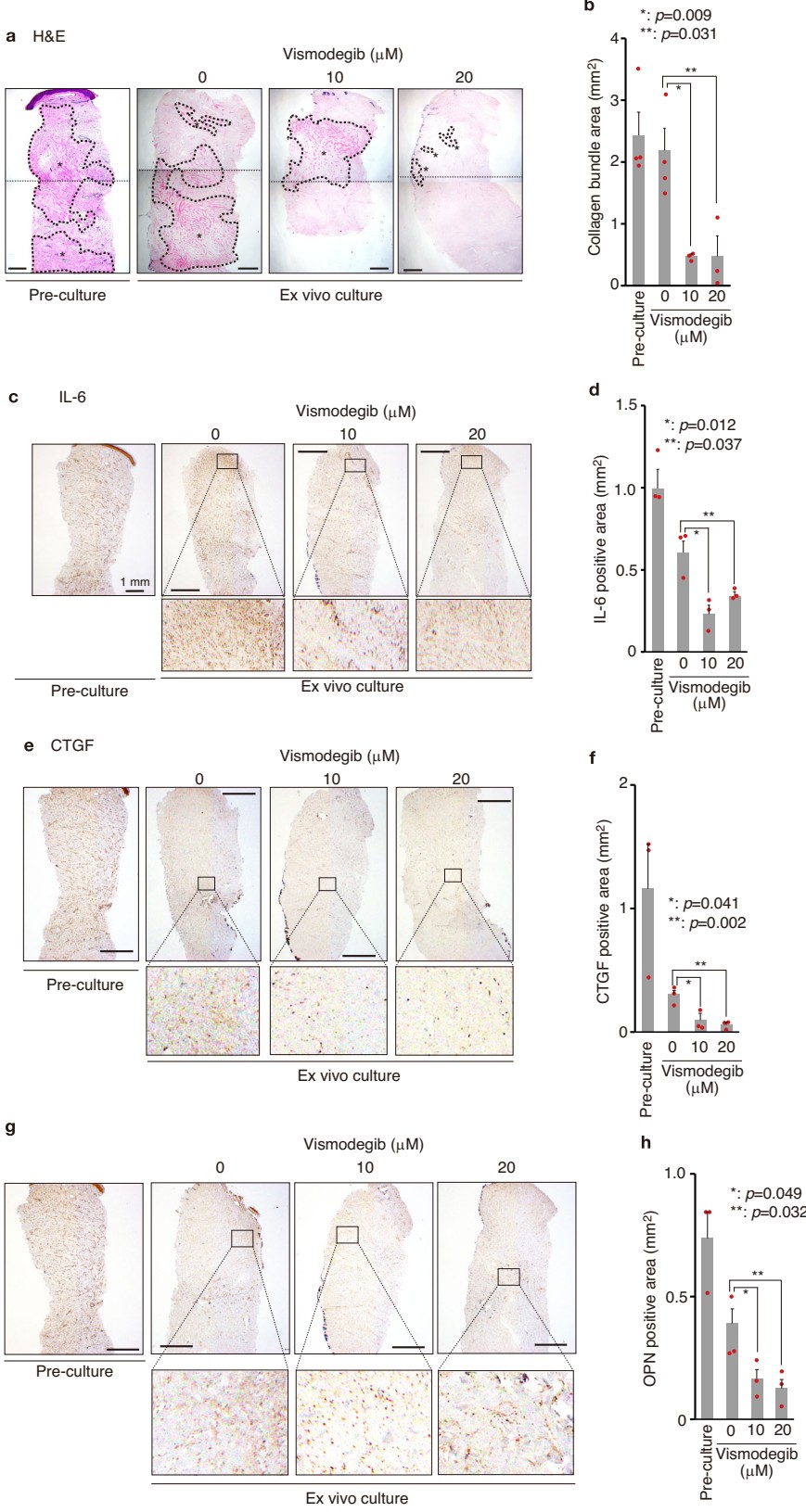

background. Moreover, genomic loci associated with microbiota diversity in chronic wounds have been identified[65]. Thus, microbiota may also be involved in the differences in HH production.

Although the mechanism of SHH production is not clear, it is possible that SHH, which functions as a morphogen and is also involved in cancer stem cell generation[40], alters the properties of stem-like cells in dermis and the altered stem-like cells may be central to keloid properties, such as induction of fibrosis and proliferation. Because GLI1-inducing signals were upregulated in three different keloid-derived stem-like cells (KNS, KMS and KCS), SHH may be a major inducer of GLI1 in keloids.

**Fig. 7 Keloid-associated factors are suppressed by inhibiting the HH signaling pathway in keloid organ culture. a, b** Inhibition of the HH signaling pathway reduced collagen bundle. Keloid tissue biopsies were taken using 4 mm diameter punch biopsy size. The biopsy sample was immediately fixed with 10% formaldehyde (Pre-culture). A schematic diagram of keloid tissue culture is shown in Supplementary Fig. 8a. Vismodegib was treated for 7 days at the indicated dose. Representative H&E staining of keloid organ culture is shown in (**a**). Bar, 500 μm. The asterisk shows the collagen bundle area. In (**b**), the keloid bundle area was quantified from four patients (K22, K23, K24, K25 [n = 4]), and the dot represents each collagen bundle area from each patient-derived keloid organ culture. Data are shown as the mean ± S.E.M. **c–h** Keloid-associated molecules were reduced by inhibiting the HH signaling pathway. Representative immunohistochemical staining is shown in (**c**) (IL-6), (**e**) (CTGF), and (**g**) (OPN). Each protein is shown in brown; bar, 500 μm. The positive area of keloid marker proteins is shown in (**d**) (IL-6), (**f**) (CTGF), and (**h**) (OPN). Results are shown as the mean ± S.E.M. from three patients (K23, K26, K27, n = 3). The dot represents each collagen bundle area from each patient-derived keloid organ culture; bar 1000 μm. Vismodegib was treated for 7 days at the indicated dose. Whole images were used to quantify positive areas in (**b**, **d**, **f**, and **h**). Patient information is shown in Supplementary Table 1; source data are provided as a Source Data file 1.

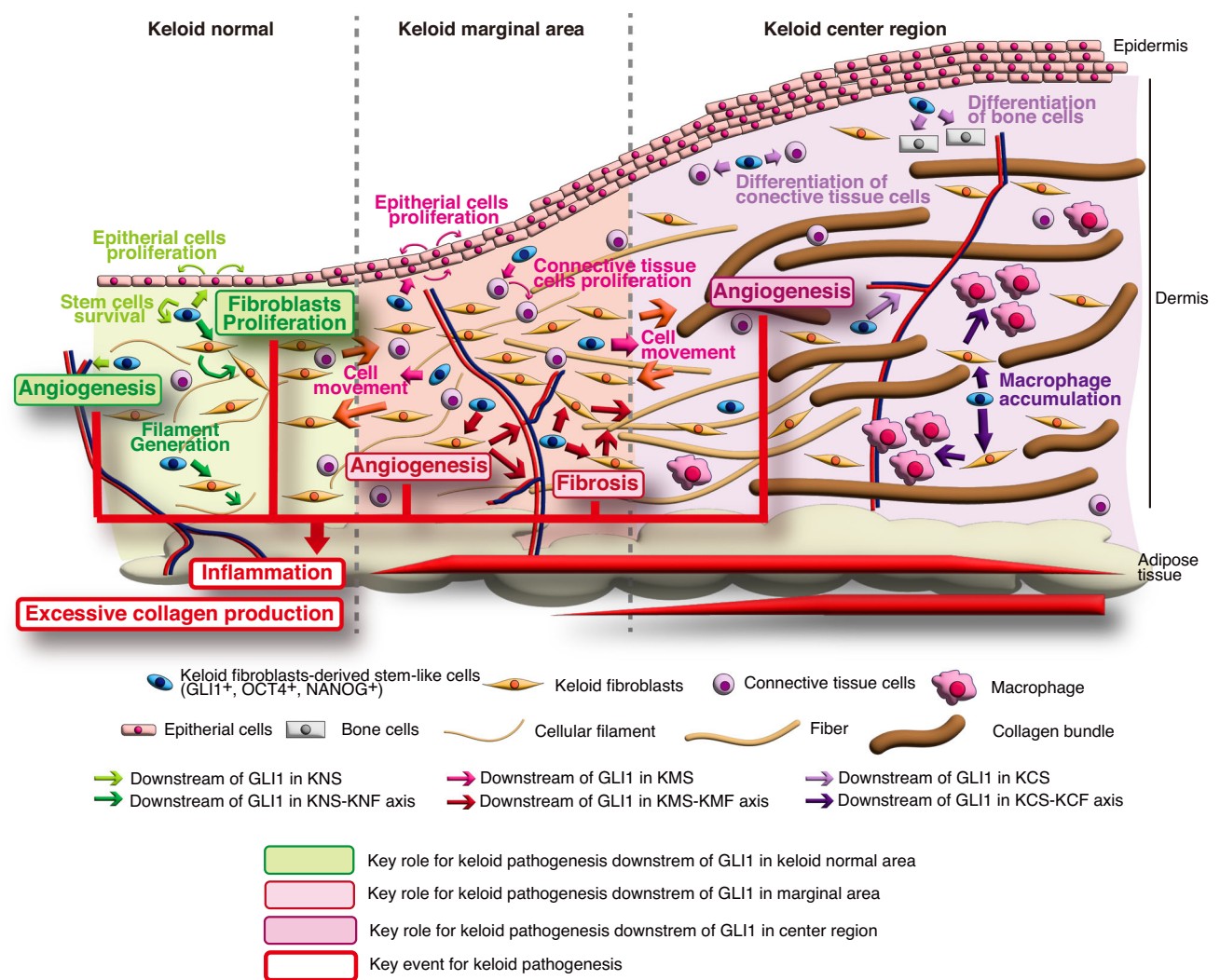

**Fig. 8 Proposed mechanisms of keloid pathogenesis via GLI1 activation.** The proposed keloid pathogenesis model through GLI1 activation in stem-like cells was generated by the pathway analysis using IPA from microarray data (Fig. 4 and Supplementary Figs. 6 and 7). The IPA results suggest that GLI1 activation in stem-like cells has a common or different role in the normal area, the marginal area, and the center region of the keloid. Also, it is suggested that GLI1-activated stem-like cells and the interaction between GLI1-activated stem-like cells and fibroblasts drive key factors for keloid pathogenesis, such as fibrosis and angiogenesis, which induce inflammation and excessive collagen production. This model suggests that the HH pathway inhibitor may have the potential to be a new treatment for the keloid.

Inhibition of the SHH signal transducer SMO inhibited the expression of GLI1 and its downstream genes and the pathologic phenotype of keloid lesions. In addition to SHH production, it is possible that the susceptibility to SHH is different in the stem-like cells at their position, because the expression pattern of GLI1 downstream genes was different between KNS, KMS and KCS. Given the different expression levels of SMO, these differences could be because of differential expression of signaling molecules of HH or regulators of GLI1 transcription at different position of the disease. Indeed, previous studies showed that transcriptional induction of target genes by GLI1 depends on various GLI1 protein modifications and different partner molecules[66]. We previously found that methylation of GLI1 protein by protein arginine methyltransferase 5 (PRMT5) is important for GLI1

activation by HH[67]. Thus, the expression and function of these protein modifiers and/or transcriptional cofactors of GLI1 may be different among keloid sites, and these differences may influence the expression of the keloid fibroproliferative phenotype. Indeed, the IPA results indicate that the gene expression profile varies greatly depending on the site of the keloid.

Fibrosis from excessive accumulation of ECM components results from chronic inflammatory reactions triggered by various stimuli, including infections, autoimmune reactions, and tissue damage[68]. Keloids are fibroproliferative tumors characterized by chronic inflammation with accumulation of ECM components, especially collagen, that may be caused by overexpression of growth factors and cytokines[10]. TGF-β, EGF and PDGF are thought to be involved in keloid fibrosis; however, their precise roles and the upstream signals that induce these cytokines have not been elucidated. In the present study, we found that the expressions of OPN, FGF1 and IGFBP7 were upregulated in keloid-derived stem-like cells in a HH-GLI1 pathway-dependent manner, and the characteristic of keloid, collagen bundle formation, and the expression of inflammatory cytokine IL-6 and fibrosis-inducing factor CTGF was suppressed by the HH signal inhibitor in human keloid tissues. Aberrant activation of the HH signaling pathway induces production of fibrosis-promoting cytokines and OPN in NKT cells, resulting in promotion of liver fibrosis[69]. While we could not fully clarify the precise mechanism of keloid fibrosis using only our experimental systems, the results of cultured cells, the in vivo reconstitution system, and ex vivo tissue culture system indicate that fibrosis may be regulated by the HH-GLI1 pathway.

There are many treatment options for keloids, including surgery, radiation, corticosteroids, cryotherapy, laser therapy, anti-allergic agents, and anti-inflammatory drugs[70]. While these treatments are improving the outcome of keloids, refractory cases and recurrence remain problems. In the present study, we found that the HH signal inhibitor vismodegib efficiently suppressed keloid-specific gene expression in an in vivo reconstitution system and keloid-specific protein expression, including keloidal collagen bundles, in an ex vivo tissue culture system. Because the lack of appropriate animal models makes it difficult to determine the therapeutic effect of treatments on keloid, we analyzed various experimental wound healing models using human tissues[71]. The results suggest that HH signaling inhibitors have potential as a new treatment for keloids. While long-term systemic administration of HH signal inhibitors for cancer is hampered by various adverse effects, including dysgeusia[72], it may be effective in the treatment of keloids, where local administration is possible. Furthermore, our present study indicated that keloid-specific changes of stem-like cells occur in the normal-looking peri-keloid area and the proliferating keloid marginal area. Therefore, a detailed study of the area of administration around the keloid and other details may help lead to more effective treatment.

## Methods

**Cell isolation and culture.** The experiments using keloid patient-derived cells were approved by the Medical and Ethics Committee of Nippon Medical School Hospital (B-2021-368). All patients signed informed consent prior to enrolling in this study. All patients in this study were Japanese. Keloid tissue was collected during keloid surgical treatment from 29 patients who were confirmed to have clinical evidence of keloids (Supplementary Table 1). No patients received chemotherapy, radiotherapy, or intralesional steroid treatment before surgery. As controls, normal skin and mature scar tissue were collected from six patients who had undergone surgical treatment other than keloids and had healed to normal scar after surgery, and six cases of normal skin and a mature scar were used for analysis. Keloids and normal skin and scars were diagnosed on the basis of their clinical appearance and pathology. Primary fibroblast cultures were established as previously described[73]. Briefly, the surgically resected keloid tissue and the normal skin tissue used only the dermis layer by removing the epidermis and fat. The dermal tissue was washed with PBS, cut into pieces ~1 mm in size, and cultured in Dulbecco's modified eagle medium (DMEM) containing 4.5 mg/ml glucose (DMEM hi-glucose: FUJIFILM Wako Pure Chemicals, Osaka, Japan) supplemented with 10% fetal bovine serum (FBS: ThermoFisher Scientific [Gibco]: Waltham, MA USA) and penicillin-streptomycin mixture (Nacalai Tesque, Kyoto, Japan). Keloid fibroblasts and normal fibroblasts were cultured in the same condition. When fibroblasts reached ~80% confluence, cells were passed at 1/3. Only low-passage cultures (passages 1 or 2) were used in this study. For keloid-derived sphere culture, fibroblasts were seeded in 6-well ultra-low attachment plates in DMEM/F-12 (3:1) (Gibco) supplemented with B-27 supplement (Gibco), 40 ng/ml basic FGF (bFGF, FUJIFILM Wako Pure Chemicals), and 20 ng/ml EGF (R&D Systems, Minneapolis, MN USA). Lenti-X 293T cells were purchased from Takara Bio (Kusatsu, Japan) and cultured in DMEM hi-glucose supplemented with 10% FBS.

**shRNA, recombinant lentivirus production and infection, and stable cell lines.** shRNAs, which generate siRNAs for target genes, were inserted into the pLKO.1 vector (Addgene, Watertown, MA USA). The target sequences are listed in Supplementary Table 2. Recombinant lentiviruses were produced in Lenti-X293T packaging cells by transient transfection with pLKO.1 plasmids and Lentiviral High Titer Packaging Mix (Takara Bio) using TransIT-Lenti transfection reagent (Mirus Bio, Madison, WI USA). Twenty-four hours after transfection, the medium was changed, and cells were incubated for an additional 24 h. Medium containing lentivirus was collected and debris was removed using a syringe filter (pore size 0.45 μm; CORNING, Corning, NY USA). The filtered culture media was supplemented with 8 μg/ml polybrene and used to infect cells. Stable shRNA-expressing cells were selected by puromycin (2.5 μg/ml: Sigma-Aldrich, Saint Louis, MO).

**RNA isolation and qPCR.** Total RNA was isolated using Nucleospin RNA (Takara Bio). RNA concentration and purity were assessed by a NanoDrop spectrophotometer (Cytiva, Marlborough, MA USA). An equal amount of total RNA was reverse transcribed by the PrimeScript RT reagent kit: Perfect Real Time (Takara Bio). cDNA was subjected to qPCR using LUNA real-time PCR Mastermix (New England Biolabs, Ipswich, MA USA) and qPCR probes (Thermo Fisher Scientific) with the StepOne-Plus Real-Time PCR System (Thermo Fisher Scientific). The qPCR probes are listed in Supplementary Table 3. Each amplification reaction was performed in triplicate. The mean and S.D. of three threshold cycles were used to calculate the amount of transcript in the sample (StepOne software v2.2.3; Thermo Fisher Scientific). Quantification of mRNA was represented in arbitrary units as the ratio of the sample quantity to the calibrator or to the mean values of control samples. All values were normalized to mRNA levels of human β-actin, the endogenous control.

**Microarray assay and pathway analysis.** RNA isolation from normal fibroblasts, normal fibroblast-derived stem-like cells, keloid fibroblasts, and keloid fibroblast-derived stem-like cells was described above. Total RNA was isolated from three keloid patients and two donors. Microarray assay was outsourced to a contracted analysis service from Takara Bio. Briefly, RNA

integrity was measured in the Bioanalyzer 2100 System (Agilent Technologies, Santa Clara, CA USA), and samples with an RNA integrity number ≥1.6 were subjected to microarray analysis. Total RNA (100 ng) was used to generate ss-cDNA, which was hybridized on the Human Clariom S Assay gene chip (Affymetrix). To investigate pathways associated with differentially expressed genes, upregulated or downregulated genes with a fold change of >1.3 ($p < 0.2$) in KNS, KMS, or KCS compared with NS were analyzed by Ingenuity Pathway Analysis software (QIAGEN, Venlo, The Netherlands).

**In vitro drug treatment and quantification of viable cells**. To evaluate the influence of SMO inhibitor on fibroblasts, $3 \times 10^3$ cells were seeded in each well of 96-well plates. When cells reached 100% confluence, the growth medium was changed to DMEM hi-glucose supplemented with 0.2% FBS, and vismodegib (Selleck, Houston, TX USA) was treated for 72 h. To evaluate the effect of vismodegib on fibroblast-derived stem-like cells, $1 \times 10^4$ cells were seeded in each well of 96-well ultra-low attachment plates, and vismodegib was treated for 7 days. The quantification of viable cells was performed using a CCK-8 staining kit (Dojindo, Kumamoto Japan) following the manufacturer's instructions.

**Immunohistochemistry**. Tissues were fixed in 10% formaldehyde for 48 h and embedded in paraffin. Samples were sectioned (4 µm) and mounted onto glass slides. For visualizing collagen bundles, tissue slides were deparaffinized and subjected to hematoxylin and eosin (H&E) staining as described elsewhere[74]. For immunohistochemical analysis, deparaffinized tissue slides were subjected to antigen retrieval by the antibody manufacturers' recommended method. Endogenous peroxidase was blocked with 0.3% hydrogen peroxide for 30 min. Nonspecific binding sites in tissues were blocked with 1% bovine serum albumin (BSA). The tissue samples were incubated with the appropriate dilution of a primary antibody overnight at 4 °C. The primary antibodies used in this study are listed in Supplementary Table 4. Biotin-conjugated secondary antibodies (described in Supplementary Table 5) were incubated for 30 min at room temperature, and interaction of the antigen and the primary antibody was detected with the streptavidin-biotin-peroxidase staining kit (Vector Laboratories, CA USA) and chromogen 3,3′-diaminobenzidine (DAB). Nuclei were stained by hematoxylin. Negative controls included the omission of primary antibody and the substitution with nonimmune sera. Adobe Photoshop 2022 (Adobe, San Jose, CA USA) was used for image analysis.

**Immunofluorescent staining**. Keloid fibroblasts seeded on cover glass slips were fixed with 4% paraformaldehyde and permeabilized with 0.5% Triton X-100. The cells were incubated with 1% BSA for 15 min at room temperature, followed by incubation with primary antibody for 30 min at room temperature. Texas red–conjugated anti-mouse IgG antibody (Thermo Fisher Scientific [Invitrogen]) was incubated for 20 min at room temperature in the absence of light. Nuclei were stained with Hoechst33342 (Invitrogen), and the coverslip was mounted onto a ProLong glass slide (Invitrogen). Cells were visualized by BioZERO fluorescence microscopy (KEYENCE, Osaka Japan).

Keloid tissues were also subjected to immunofluorescence staining. Slices of frozen sections (4 µm-thick) were mounted on a slide glass and dried for 30 min at room temperature. The tissue was fixed in 4% paraformaldehyde for 10 min at room temperature and incubated with 1% BSA for 30 min at room temperature. The tissue was then incubated with anti-GLI1 and anti-OCT4 or anti-NANOG antibodies overnight at 4 °C. The

sections were incubated with Alexa 488-conjugated anti-mouse (for GLI1) and Alexa 546-conjugated anti-rabbit (for OCT and NANOG) secondary antibodies for 30 min at room temperature, avoiding light. Nuclei were stained with VECTASHILD® Antifade Mounting Medium with 4′,6′-diamidino-2-phenylindole (DAPI: Vector Laboratories, Burlingame, CA USA), and the tissue was covered with cover glass. Immunofluorescent images were visualized by XA70 fluorescence microscopy (EVIDENT [Olympus], Tokyo Japan). Antibodies used in the immunofluorescent staining are listed in Supplementary Tables 4 (Primary antibodies) and 5 (Secondary antibodies).

**Flow cytometric analysis**. In total, $5 \times 10^5$ (KNS) or $1 \times 10^6$ cells (NF, NS, KNF) were used for flow cytometric analysis. Fibroblasts were rinsed with ice-cold PBS and dissociated with 2.5% Trypsin solution (Nacalai Tesque). For the dissociation of sphere-forming cells, Accumax solution (Nacalai Tesque) was used after rinsing the sphere. After the dissociation of cells, cells were washed with ice-cold FCM buffer (2% FBS in PBS) twice. Then, cells were stained with isotype control and the cell surface marker (described in Supplementary Table 4), followed by treatment of Human TruStain FcM™ (Fc Receptor Blocking Solution: BioLegend, San Diego, CA USA). Labeled cells were subjected to Cytoflex S flow cytometer (BECKMAN COULTER, Brea, CA USA). The result was analyzed by FlowJo ver. 10 software (Becton Dickinson [BD], Franklin Lakes, NJ USA).

**In vivo keloid fibroblast transplantation and gene set enrichment analysis (GSEA)**. The animal experiment committee at the Nippon Medical School approved the animal experiments in this study (approval number: 29-045), and animal care was conducted by institutional guidelines. In vivo transplantation was performed as described elsewhere[35]. Briefly, $2 \times 10^6$ keloid fibroblasts from the marginal area (KMF) from three patients or normal fibroblasts from two donors were mixed with hydrogel including synthesized extracellular matrix (HyStem®-HP Cell Culture Scaffold Kit, Sigma-Ardrich) supplemented with recombinant human IL-6 (50 ng/ml: PeproTech, Cranbury, NJ USA). The mixtures were immediately injected subcutaneously into the dorsal surface of 6–8-week-old male nude mice (Japan CREA, Tokyo, Japan). KMF-transplanted mice were treated with vehicle or vismodegib by intraperitoneal injection once a day. Mice were killed 3 months after implantation, and the harvested transplant volume was evaluated. Transplant volume (mm$^3$) was measured with calipers and was calculated as (length × width$^2$)/2. RNA was isolated from the transplant for qPCR analysis and microarray assay. For GSEA of microarray assay data, G5:GO gene sets (The Molecular Signature Database [www.broadinstitute.org/gsea/msigdb]) were used for the enrichment of keloid pathogenesis using GSEA software V3.0 (Broad Institute, MIT).

**Ex vivo keloid tissue culture**. Ex vivo keloid tissue culture was performed as described elsewhere[56]. Keloid tissue culture media was composed of William's E (WE) medium (Merck) supplemented with 10 µg/ml of recombinant human insulin (FUJIFILM Wako Pure Chemicals), 10 ng/ml of hydrocortisone (FUJIFILM Wako Pure Chemicals), 2 mM of L-glutamine (FUJIFILM Wako Pure Chemicals), and antibiotics. The keloid tissue biopsies were taken using a 4 mm diameter punch biopsy size and embedded in keloid tissue culture media supplemented with calf collagen matrix (3 mg/ml: KOKEN, Tokyo, Japan) in 24-well plates. Keloid tissue culture media was added to the culture plate to expose the surface of the epidermis to the air. For tissue maintenance, the media was changed every 3 days and supplemented with fresh media.

**Statistics and reproducibility**. The statistical significance of the difference between mean values was tested using a two-tailed Student's *t* test with unequal variance in Microsoft Excel. The significance threshold was set at $p \leq 0.05$, except for microarray analysis. *p* values are indicated in each figure. The representative immunofluorescent images in Fig. 2d, f and Supplementary Fig. 3b, c are from five independent experiments from five keloid patients (K8, K9, K10, K11, K12) and three independent experiments from three donors with normal dermis (N3, N4, N5). The representative immunohistochemical images in Figs. 3a and 4f are from five independent experiments from five keloid patients (K1, K2, K3, K4, K9). The representative H&E staining image in Supplementary Fig. 7 of keloid fibroblasts transplants is from three keloid patient–derived fibroblasts (K12, K13, K18) and two donors with normal dermis–derived fibroblasts (N3, N4). The representative ex vivo histology images in Fig. 7 are from four independent experiments from four keloid patients (K19, K20, K21, K22) per condition. The representative ex vivo histology images in Supplementary Fig. 8 are from two independent experiments from keloid patients (K23, K24) per condition. The representative image in Supplementary Fig. 2a is from five (POU5F1 [K2, K3, K4, K5 and K6] and NANOG [[K2, K3, K4, K5 and K6]] or six (SOX2 [K1, K2, K3, K4, K5 and K6] independent experiments. The representative image in Supplementary Fig. 5b is from four (K1, K3, K7, K10) independent experiments. The representative image in Supplementary Fig. 9c is from two (K22 and K23) independent experiments.

**Reporting summary**. Further information on research design is available in the Nature Portfolio Reporting Summary linked to this article.

## Data availability

Microarray data for pathway analysis were deposited in the Gene Expression Omnibus (GEO) database under accession number GSE218922. Microarray data for GSEA were also deposited in the GEO database under accession number GSE218894. All other data are available in the article and its Supplementary Information and Source Data file or from the corresponding author upon reasonable request. Source data are provided with this paper.

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

## Acknowledgements

We thank Akihiro Yamazaki, Naoki Otsuka, and Masafumi Ogawa for their technical assistance. We thank Takenori Fujii for assistance with immunohistochemical analysis. We also thank Gabrielle White Wolf, PhD from Edanz Group (https://jp.edanz.com/ac) for editing a draft of this manuscript. This work was supported by JSPS KAKENHI Grant Number 17K11557.

## Author contributions

M.T. and Y.A. analyzed and interpreted all data and developed the study concept. N.T. and R.O. assisted study concept creation. M.T. obtained funding and collected patient-derived samples after the keloid surgical operation. S.E. conducted histological experiments. M.T., T.H., C.I., A.M., R.M., and T.S. conducted qPCR experiments to validate microarray results and GLI1 downstream pathway analysis. T.H. performed pathway analysis using qPCR data. Y.A. conducted other investigations and analyses in this study. N.T., R.O., M.T., and Y.A. drafted and edited the manuscript. All authors approved the final version of the manuscript.

## Competing interests

The authors declare no competing interests.
