## [Peer Review File · Communications Biology]

Reviewers' comments:

Reviewer #1 (Remarks to the Author):

These investigators compared the characteristics of stem cells from keloid lesions and the surrounding dermis to those of normal skin. HEDGEHOG (HH) signal and its downstream transcription factor GLI1 were upregulated in keloid patient-derived stem cells. Inhibition of the HH-GLI1 pathway reduced the expression of genes involved in keloids and fibrosis-inducing cytokines. The HH signal inhibitor vismodegib reduced keloid reconstituted tumor size and keloid-related gene expression in nude mice and the collagen bundle and expression of cytokines characteristic for keloids in ex vivo culture of keloid tissues. Their results implicate the HH-GLI1 pathway in keloid pathogenesis and suggest therapeutic targets of keloids.

Overall, the manuscript is very well written, contains large amounts of high quality data in 7 figures and 7 supplementary figures as well as tables and data files. The authors have employed in vitro cell cultures experiments, a novel impressive keloid model in vivo as well as ex vivo techniques to establish their findings.

Concerns with the Manuscript:

1. Can the authors provide evidence to support the description of stem cells derived from keloids as being truly pluripotent stem cells by definition?
2. In figure 6 b, considerable variability exists in the data displayed and the magnitude of the effect of Vismodegib while statistically significant is relatively small.
3. The ex vivo experiments present single images of keloid organ cultures which are prone to sampling error depending on the section selected for staining. How did the authors avoid this in their experiments?
4. Have the stem cells from keloids been characterized to assist in understanding the origin of the cells, are they bone marrow derived or resident cells?
5. Similarly, the authors allude to a key event for keloid pathogenesis however their data do not establish this.
6. SHH signalling is associated with cross talk of a number of other pathways including TGF- β and Wnt signalling pathways which have been implicated in fibrosis. To what extent could their findings with SHH be independent or involved with these well recognized profibrotic molecules?
7. What and how is a normal fibroblast stem-like cell characterized and then used for comparison in the supplementary figure 6?
8. Although of lesser importance, supplementary figure 4 is poorly focussed.
9. In supplementary figure 6 part d the label KPS appears for one of the sets of cells?
10. p4 l78 wound hearing should be wound healing?

Reviewer #2 (Remarks to the Author):

The manuscript describes a putative role for SHH/GLI signaling in keloid pathogenesis based on differential expression in normal and keloid fibroblasts and "stem cells." Additionally, the study explores in preclinical models the potential role for the SHH inhibitor vismodegib as a therapeutic approach to reducing keloid volume. Keloids are a challenging problem, and more effective therapies are needed. While this is an important area of inquiry, I have major concerns regarding the terminology used here to describe some fibroblasts as "stem cells" based on a method of culturing primary fibroblasts that was purported to select for stem cells. By culturing primary fibroblasts under "neural sphere-forming conditions," these cells are apparently turned into a uniform population of stem cells?

The authors initially use the term "keloid stem-like cells" in the Introduction, but switch to the term "stem cells" in the Results and Discussion. From the descriptions provided, their study involved primary fibroblasts, but not stem cells. The methods describe isolation of fibroblasts from dermis of normal skin or keloids by explant culture (ie, 1 mm size pieces were cultured in DMEM, and cells were passaged 1:3 upon reaching 80% confluence). In a later section of the Methods describing the microarray assay and pathway analysis, they say they isolated RNA from "normal fibroblasts, normal fibroblast-derived stem cells, keloid fibroblasts, and keloid fibroblast-derived stem cells," but nowhere do they explain how "stem cells" were isolated, or how they were confirmed to be stem cells. Are the authors assuming that simply by virtue of spheroid culture, they have isolated stem cells, or converted primary cells to stem cells? Do they know that these are all stem cells? Based on what criteria?

It is not surprising that cells grown in monolayer vs cells grown in spheroid culture, with different basal media and different sets of growth factors added, would display different gene expression profiles (whether or not one group contains stem cells). However, I am not convinced that the cells cultured in spheroid culture can all be considered stem cells, or that none of the cells in fibroblast primary monolayer cultures are stem cells. They provide a citation for this (#35), but I do not believe it is widely accepted in the field that sphere-forming ability is "a standard method for determining the stem cell population," as the authors assert. In the paper, the authors assume that all gene expression differences are due to "mature fibroblasts" vs "stem cells" differences, but differences could simply be due to differences in culture conditions: the fibroblasts in monolayer were cultured in DMEM + 10% FBS, whereas the spheroid "stem cell" cultures were grown in DMEM/F12 with B27 supplement (which contains numerous vitamins, hormones, and other factors), bFGF and EGF. "Mature fibroblasts" isn't really even an accurate description of primary cultured fibroblasts, as the process of establishing a primary culture causes fibroblast activation (and changes the gene expression profile vs. cells in vivo). Isn't it possible that expression differences are due to the differences in media and growth factors used for culture? (note, for example, that both EGF and bFGF have been shown to regulate expression of SHH pathway genes, as there is significant crosstalk among these signaling pathways) One way to resolve this issue is to see if keloid fibroblasts grown in standard monolayer culture but using the spheroid culture medium formulation (with B27, bFGF and EGF) showed the same gene expression pattern as the spheroids...this could answer the question of whether the difference in SHH pathway gene expression is due to the culture media or some trait specific to this purported "stem cell" population, and would be an important control to address differences in culture media as a confounding factor. Additionally, analysis of putative stem cell markers in the keloid and normal "stem cell" populations would help address this concern, although this could still be an artifact of in vitro culture with uncertain relevance to tissue in vivo.

Given the uncertainty regarding the stem cell identity of the in vitro spheroid culture-derived "keloid stem cells," the title should be revised; although the current study, and previous publications, strongly suggest a role for SHH signaling based on differential gene expression (and the role of this pathway in skin physiology and wound healing), the current study does not demonstrate that "keloid pathogenesis and fibroproliferative properties are dependent on stem cells regulated by the hedgehog-Gli1 pathway." The data are suggestive, and support a role for SHH/GLI1 as a therapeutic target, but the role of any putative stem cell population in keloid pathogenesis in vivo is not unequivocally demonstrated here.

Other concerns are listed below.

In the first section of the Results, the authors describe the "central regions" of keloids as "shrunken and soft, termed the "older parts of the keloids."" This is not strictly true, as many keloids (particularly the bulbous types) have very raised, hard centers. In this same paragraph, the authors describe

differences in gene expression signatures of different cell populations; these descriptions are confusing as presented.

The authors have included a great deal of background information in the Results section, which is a bit unusual; this material really belongs in the Introduction (or Discussion, as relevant), whereas the Results section should be limited to just reporting the results of experiments. Otherwise, it may be difficult for readers to understand what was done in the current study vs what was previously reported (even if citations are included).

In the Methods, the authors first describe collection of keloids from 27 patients, and then subsequently describe that "normal skin and scar tissue were obtained from four patients who underwent elective scar excision surgery." Presumably, these "scar" patients are distinct from the keloid patients, but there is a discrepancy between this information in the text and Table 1, which lists 5 normal skin samples and one mature scar sample. This is confusing and should be clarified. Further, it is unclear how many primary cell cultures were established and how many distinct donor-derived primary cultures were utilized in experiments (I see that this information is in the statistical analysis section, but the information should be easier to find...see additional comment, below).

The authors provide a table describing the keloids and normal skin samples used in the study (Supplementary Table 1). In the statistical analysis section of the Methods, they describe which samples were used for which studies. It is not clear why different samples (and different numbers of samples) were used for different parts of the study. This makes sense in some cases (eg, ex vivo cultures), though if histological sections were prepared, it is not clear why immunohistochemistry could not be performed on all tissue samples. Did the authors perform a power analysis to determine that N=5 for keloids and N=3 for normal samples is sufficient for statistical analysis? There is information in Table 1 that is the same for all donors and thus does not need to be listed; for example, all donors were Japanese and none had medication pre-surgery. Instead, it would be useful to include in this table which donor samples were used for which experiments (making it easier for readers to determine how representative those samples really were—for example, were only chest keloids examined in some cases?). There can be great variability among donors and among different keloids, so it is important to understand which samples were analyzed at which points in the study. The authors have provided the information but it is currently tedious to tease out the answers. Adding this information to Table 1 would simplify this issue.

Responses to the reviewers' comments

MS: COMMSBIO-23-1553-T

Title: Keloid pathogenesis and fibroproliferative properties are dependent on stem cells regulated by the HEDGEHOG-GLI1 pathway

Authors: Mamiko Tosa, Yoshinori Abe, Seiko Egawa, Tomoka Hatakeyama, Chihiro Iwaguro, Ryotaro Mitsugi, Ayaka Moriyama, Takumi Sano, Rei Ogawa, Nobuyuki Tanaka

We are grateful to the referees for their invaluable comments and suggestions. In accordance with their suggestions, we performed additional experiments so as to adequately address the issues raised and have amended the paper accordingly. Please find below our point-by-point responses to each of the comments.

Reviewer #1:

Concerns with the Manuscript:

1. Can the authors provide evidence to support the description of stem cells derived from keloids as being truly pluripotent stem cells by definition?

This is the main concern of reviewer #2 and we agree that this needs to be clarified. Therefore, we performed additional experiments, but we could not definitively conclude that the sphere-forming cells were stem cells. Therefore, we have changed all “stem cell” phrases to “stem-like cells,” and have added the following results to the text: “We examined the expression of reprogramming factors OCT4 and NANOG, which regulate a cascade of pathways to control pluripotency, self-renewal, genome surveillance, and cell fate determination⁴⁴. As shown in Supplementary Fig. 1a–d, we found that the expression of OCT4 and NANOG proteins was not detected in fibroblasts in and around the keloid sites, and in normal fibroblasts, but most of the sphere-forming cells in these sites expressed these proteins at high levels. In contrast, the expression of OCT4 and NANOG proteins was increased in sphere-forming cells, while mRNA expression was induced, albeit not as significantly as these proteins (Supplementary Fig. 1e). This may have been due to the finding that the expression of these reprogramming factors in stem cells is mainly regulated by post-transcriptional modifications⁴⁵. Moreover, we analyzed the gene expression profile in sphere-forming cells by

microarray analysis and identified upregulation of the signaling pathways for human embryonic stem cell pluripotency using Ingenuity Pathway Analysis (IPA) software (Supplementary Fig. 1f). These results suggest that the sphere-forming cells are homogeneous and have stem cell-like characteristics. However, we did not find any stem cell markers previously reported in several experimental systems^{35, 46} that are specifically increased in sphere-forming cells (Supplementary Fig. 1g-l). These results did not lead to the conclusion that the sphere-forming cells were stem cells, but suggested that they were stem cell-like cells with a stem cell phenotype” (lines 114 to 130).

2. In figure 6 b, considerable variability exists in the data displayed and the magnitude of the effect of Vismodegib while statistically significant is relatively small.

As pointed out by the reviewer, there was considerable variability in the data shown in Figure 6b. However, in our later experiments, we showed in various analyses the effect of vismodegib on keloids. We thought that it would be fairer to the readers to describe these. Therefore, we have added the following description to the text: “Meanwhile, although these *in vivo* results are statistically significant, they have considerable variability, making it difficult to estimate the magnitude of the effect of vismodegib. Therefore, we used various analyses and experiments to examine the effects of vismodegib on keloids” (lines 230 to 233).

3. The ex vivo experiments present single images of keloid organ cultures which are prone to sampling error depending on the section selected for staining. How did the authors avoid this in their experiments?

We agree with this comment. To avoid concerns about sampling errors, we have shown in Supplementary Figure 10 the entire tissue staining of all different patient specimens that we used.

4. Have the stem cells from keloids been characterized to assist in understanding the origin of the cells, are they bone marrow derived or resident cells?

The lack of a mouse model for keloids has made it difficult for us to perform experiments such as bone marrow transplantation and cell fate mapping. Furthermore, even with surface marker analysis, we were unable to estimate the origin.

5. Similarly, the authors allude to a key event for keloid pathogenesis however their data do not establish this.

Indeed, we did not reveal the mechanism by which SHH expression is increased in keloid tissues and the mechanism by which activation of SHH signaling alters properties of stem-like cells. Although we accept that a detailed explanation is lacking, we believe that our results suggest that the HH-GLI1 pathway is involved in the pathogenesis of keloids. Therefore, we have added the following description at the beginning of the Discussion: “In the present study, we found that the characteristics of stem-like cells from keloid lesions and surrounding dermis differed from those of normal skin, with increased activity of HH signaling and its downstream transcription factor GLI1. We also found that SHH protein is widely expressed in keloids and surrounding dermal tissues. Moreover, we found that inhibition of the HH-GLI1 pathway decreased the number of keloid stem-like cells in cultured keloid fibroblasts and *in vivo* reconstructed keloid tissue in nude mice, and decreased the expression of genes involved in keloids and fibrosis-inducing cytokines that induce collagen expression. Simultaneously, the HH signaling inhibitor vismodegib reduced reconstructed keloid tumor size and keloid-related gene expression in nude mice, and reduced collagen bundles and the expression of cytokines characteristic of keloids in *ex vivo* cultures of keloid tissue. Although we did not reveal the mechanism by which SHH expression is increased in keloid tissues and by what mechanism activation of SHH signaling alters properties of stem-like cells, our results suggest that the HH-GLI1 pathway is involved in the pathogenesis of keloids” (lines 261 to 273).

6. sHH signalling is associated with cross talk of a number of other pathways including TGF- β and Wnt signalling pathways which have been implicated in fibrosis. To what extent could their findings with sHH be independent or involved with these well recognized profibrotic molecules?

We agree with this comment and have performed additional experiments accordingly. We have added the obtained results in fig. S10 and S12, along with the following descriptions in the text: “HH signaling is known to crosstalk with TGF- β and WNT signaling pathways that are involved in fibrosis⁵⁷. Therefore, we investigated whether the enhanced HH signaling observed in keloid tissues affects well-known pro-fibrotic molecules. However, as shown in Supplementary Fig. 10d–f, the expression of the WNT-signaling molecule β -Catenin, TGF- β , and its effector transcription factor SMAD3 in keloid tissue was not significantly altered among tissues derived from four patients” (lines 246 to 251), and “We also found that the expression of TGF- β , but not β -Catenin, was suppressed (Supplementary Fig. 12a, b), suggesting that TGF- β signaling is activated downstream of HH signaling” (lines 253 to 255).

7. What and how is a normal fibroblast stem-like cell characterized and then used for comparison in the supplementary figure 6?

This point was also based on our lack of explanation. Accordingly, we have added the following description to the legend of Supplementary Fig. 7g: “KNS, KMS and KCS was described in Fig. 1.”

8. Although of lesser importance, supplementary figure 4 is poorly focussed.

We believe that the reviewer was concerned by our failure to explain this figure. We apologize for this and have added the following description to the text: “Because GLI1 is a target gene of the HH signal pathway³⁹, to analyze the expression of SHH in keloid tissue, we performed immunocytochemistry, which showed enhanced protein expression of SHH in keloid tissue and surrounding normal-looking tissues compared with normal skin and mature scar from a non-keloid patient (Fig. 3a–d) and SHH was mainly expressed in the cytoplasm (Supplementary Fig. 5)” (lines 171 to 175).

9. In supplementary figure 6 part d the label KPS appears for one of the sets of cells?

This was a typo. We corrected “KPS” to “KMS.”

10. p4 178 wound hearing should be wound healing?

This was also a typo. We corrected “hearing” to “healing” (line 75).

Reviewer #2:

The manuscript describes a putative role for SHH/GLI signaling in keloid pathogenesis based on differential expression in normal and keloid fibroblasts and “stem cells.” Additionally, the study explores in preclinical models the potential role for the SHH inhibitor vismodegib as a therapeutic approach to reducing keloid volume. Keloids are a challenging problem, and more effective therapies are needed.

While this is an important area of inquiry, I have major concerns regarding the terminology used here to describe some fibroblasts as “stem cells” based on a method of culturing primary fibroblasts that was purported to select for stem cells. By culturing primary

fibroblasts under “neural sphere-forming conditions,” these cells are apparently turned into a uniform population of stem cells?

In accordance with this comment, we examined the expression of reprogramming factors OCT4 and NANOG, which regulate a cascade of pathways to control pluripotency, self-renewal, genome surveillance, and cell fate determination. As shown in Supplementary Fig. 1a–d, we did not detect the expression of OCT4 and NANOG proteins in fibroblasts in and around the keloid sites, but most of the sphere-forming cells in these sites expressed these proteins at high levels. These results suggest that the sphere-forming cells are homogeneous and have stem cell characteristics. Of course, we cannot conclude from these results alone that these sphere-forming cells are stem cells.

The authors initially use the term “keloid stem-like cells” in the Introduction, but switch to the term “stem cells” in the Results and Discussion. From the descriptions provided, their study involved primary fibroblasts, but not stem cells. The methods describe isolation of fibroblasts from dermis of normal skin or keloids by explant culture (ie, 1 mm size pieces were cultured in DMEM, and cells were passaged 1:3 upon reaching 80% confluence). In a later section of the Methods describing the microarray assay and pathway analysis, they say they isolated RNA from “normal fibroblasts, normal fibroblast–derived stem cells, keloid fibroblasts, and keloid fibroblast–derived stem cells,” but nowhere do they explain how “stem cells” were isolated, or how they were confirmed to be stem cells. Are the authors assuming that simply by virtue of spheroid culture, they have isolated stem cells, or converted primary cells to stem cells? Do they know that these are all stem cells? Based on what criteria?

We agree with the reviewer’s concern. Indeed, stem cells are defined differently in various tissues, each supported by a large body of research. In contrast, few studies of keloid stem cells have been performed and their definition is not well established. Therefore, we agree that our use of the term “stem cell” is not accurate. As described above, the sphere-forming cells that we have identified express high levels of OCT4 and NANOG, and we consider that they exhibit a stem cell-like phenotype. For these reasons, we have changed all “stem cell” phrases to “stem-like cells.”

We have also described the methods for isolating sphere-forming cells in the Methods: “For the culture of keloid-derived spheres, fibroblasts were seeded in six-well ultra-low-attachment plates in DMEM/F-12 (3:1) (Gibco) supplemented with B-27 supplement (Gibco), 40 ng/ml basic FGF (bFGF, FUJIFILM Wako Pure Chemicals), and 20 ng/ml EGF (R&D Systems, Minneapolis, MN USA). Lenti-X 293T cells were purchased from Takara Bio (Kusatsu, Japan) and cultured in DMEM hi-glucose supplemented with 10% FBS” (lines 358 to 363).

It is not surprising that cells grown in monolayer vs cells grown in spheroid culture, with different basal media and different sets of growth factors added, would display different gene expression profiles (whether or not one group contains stem cells). However, I am not convinced that the cells cultured in spheroid culture can all be considered stem cells, or that none of the cells in fibroblast primary monolayer cultures are stem cells. They provide a citation for this (#35), but I do not believe it is widely accepted in the field that sphere-forming ability is “a standard method for determining the stem cell population,” as the authors assert. In the paper, the authors assume that all gene expression differences are due to “mature fibroblasts” vs “stem cells” differences, but differences could simply be due to differences in culture conditions: the fibroblasts in monolayer were cultured in DMEM + 10% FBS, whereas the spheroid “stem cell” cultures were grown in DMEM/F12 with B27 supplement (which contains numerous vitamins, hormones, and other factors), bFGF and EGF. “Mature fibroblasts” isn’t really even an accurate description of primary cultured fibroblasts, as the process of establishing a primary culture causes fibroblast activation (and changes the gene expression profile vs. cells in vivo). Isn’t it possible that expression differences are due to the differences in media and growth factors used for culture? (note, for example, that both EGF and bFGF have been shown to regulate expression of SHH pathway genes, as there is significant crosstalk among these signaling pathways) One way to resolve this issue is to see if keloid fibroblasts grown in standard monolayer culture but using the spheroid culture medium formulation (with B27, bFGF and EGF) showed the same gene expression pattern as the spheroids...this could answer the question of whether the difference in SHH pathway gene expression is due to the culture media or some trait specific to this purported “stem cell” population, and would be an important control to address differences in culture media as a confounding factor. Additionally, analysis of putative stem cell markers in the keloid and normal “stem cell” populations would help address this concern, although this could still be an artifact of in vitro culture with uncertain relevance to tissue in vivo.

We believe that these concerns are very important. We therefore examined whether the elevated expression of GLI1 mRNA observed in sphere-forming cells was not observed in fibroblasts cultured under the same conditions (Supplementary Fig. 3d). Moreover, we analyzed the gene expression profile of sphere-forming cells and found upregulation of the signaling pathways for human embryonic stem cell pluripotency (Supplementary Fig. 1e). However, we did not find any

stem cell markers previously reported in several experimental systems that are specifically increased in sphere-forming cells (Supplementary Fig. 1f–k).

Considering the above results, we have added the following descriptions to the text: “We examined the expression of reprogramming factors OCT4 and NANOG, which regulate a cascade of pathways to control pluripotency, self-renewal, genome surveillance, and cell fate determination⁴⁴. As shown in Supplementary Fig. 1a–d, we found that the expression of OCT4 and NANOG proteins was not detected in fibroblasts in and around the keloid sites, and in normal fibroblasts, but most of the sphere-forming cells in these sites expressed these proteins at high levels. In contrast, the expression of OCT4 and NANOG proteins was increased in sphere-forming cells, while mRNA expression was induced, albeit not as significantly as these proteins (Supplementary Fig. 1e). This may have been due to the finding that the expression of these reprogramming factors in stem cells is mainly regulated by post-transcriptional modifications⁴⁵. Moreover, we analyzed the gene expression profile in sphere-forming cells by microarray analysis and identified upregulation of the signaling pathways for human embryonic stem cell pluripotency using Ingenuity Pathway Analysis (IPA) software (Supplementary Fig. 1f). These results suggest that the sphere-forming cells are homogeneous and have stem cell-like characteristics. However, we did not find any stem cell markers previously reported in several experimental systems^{35, 46} that are specifically increased in sphere-forming cells (Supplementary Fig. 1g–l). These results did not lead to the conclusion that the sphere-forming cells were stem cells, but suggested that they were stem cell-like cells with a stem cell phenotype” (lines 114 to 130) and “Furthermore, the increased expression of GLI1 mRNA observed in sphere-forming cells was not observed in fibroblasts cultured under the same conditions (Supplementary Fig. 3d), suggesting that the activation of GLI1 in sphere-forming cells was not due to differences in cell culture conditions, but rather to their change into stem-like cells” (lines 162 to 165).

Given the uncertainty regarding the stem cell identity of the in vitro spheroid culture-derived “keloid stem cells,” the title should be revised; although the current study, and previous publications, strongly suggest a role for SHH signaling based on differential gene expression (and the role of this pathway in skin physiology and wound healing), the current study does not demonstrate that “keloid pathogenesis and fibroproliferative properties are dependent on stem cells regulated by the hedgehog-Gli1 pathway.” The data are suggestive, and support a role for SHH/GLI1 as a therapeutic target, but the role of any putative stem cell population in keloid pathogenesis in vivo is not unequivocally demonstrated here.

We agree with this comment and have changed the title to “The HEDGEHOG-GLI1 pathway is important for fibroproliferative properties in keloids and as a candidate therapeutic target.”

Other concerns are listed below.

In the first section of the Results, the authors describe the “central regions” of keloids as “shrunken and soft, termed the “older parts of the keloids.”” This is not strictly true, as many keloids (particularly the bulbous types) have very raised, hard centers. In this same paragraph, the authors describe differences in gene expression signatures of different cell populations; these descriptions are confusing as presented.

This was our mistake and the reviewer’s comments are correct. Therefore, we have changed this sentence as follows: “Different keloid regions exhibit different growth characteristics; the central portion of the keloid is a red, elastic, hard, raised lesion (termed the “fibrosis parts of keloid”), while the marginal (peripheral) regions tend to grow and invade into normal skin, termed the “proliferative” or “invasive” keloid regions⁴³” (lines 106 to 109).

The authors have included a great deal of background information in the Results section, which is a bit unusual; this material really belongs in the Introduction (or Discussion, as relevant), whereas the Results section should be limited to just reporting the results of experiments. Otherwise, it may be difficult for readers to understand what was done in the current study vs what was previously reported (even if citations are included).

We agree with this comment. Accordingly, we moved the description of the HH-GLI1 signal to the Introduction (lines 84 to 97).

In the Methods, the authors first describe collection of keloids from 27 patients, and then subsequently describe that “normal skin and scar tissue were obtained from four patients who underwent elective scar excision surgery.” Presumably, these “scar” patients are distinct from the keloid patients, but there is a discrepancy between this information in the text and Table 1, which lists 5 normal skin samples and one mature scar sample. This is confusing and should be clarified. Further, it is unclear how many primary cell cultures were established and how many distinct donor-derived primary cultures were utilized in experiments (I see that this information is in the statistical analysis section, but the information should be easier to find...see additional comment, below).

Thank you for pointing this out. This was our mistake, and we have made the following changes to the relevant section of Methods. The number of samples has been changed in the text and in Table 1 to the final version after additional experiments.

We have changed “Keloid tissue was collected during keloid surgical treatment from 27 patients” to “Keloid tissue was collected during keloid surgical treatment from 29 patients” (lines 343 to 344), and “Normal skin and scar tissue were obtained from four patients who underwent elective scar excision surgery” to “As controls, normal skin and mature scar tissue were collected from six patients who had undergone surgical treatment for conditions other than keloids and had healed to a normal scar after surgery, and six cases of normal skin and a mature scar were used for analysis” (lines 346 to 348).

The authors provide a table describing the keloids and normal skin samples used in the study (Supplementary Table 1). In the statistical analysis section of the Methods, they describe which samples were used for which studies. It is not clear why different samples (and different numbers of samples) were used for different parts of the study. This makes sense in some cases (eg, ex vivo cultures), though if histological sections were prepared, it is not clear why immunohistochemistry could not be performed on all tissue samples. Did the authors perform a power analysis to determine that N=5 for keloids and N=3 for normal samples is sufficient for statistical analysis?

Thank you for this comment, with which we agree. In this study, we examined three areas of keloid tissue. The reddened area around the keloid and the normal skin outside it are very limited. When used for analysis other than immunohistochemical staining, many samples did not have enough tissue left for immunohistochemical staining. Therefore, it was not possible to perform immunohistochemical staining on these samples, which is considered a limitation of the immunohistochemical analysis.

There is information in Table 1 that is the same for all donors and thus does not need to be listed; for example, all donors were Japanese and none had medication pre-surgery. Instead, it would be useful to include in this table which donor samples were used for which experiments (making it easier for readers to determine how representative those samples really were—for example, were only chest keloids examined in some cases?). There can be great variability among donors and among different keloids, so it is important to understand

which samples were analyzed at which points in the study. The authors have provided the information but it is currently tedious to tease out the answers. Adding this information to Table 1 would simplify this issue.

In accordance with this comment, we have deleted the common information and included in Supplementary Table 1 which donor samples were used for which experiments.

REVIEWERS' COMMENTS:

Reviewer #1 (Remarks to the Author):

The revised manuscript has been reviewed including figures and supplementary data and the response to the original review.

Considerable additional data have been performed to address the concerns of both reviews.

The revised manuscript has acknowledged the principal concern of lack of rigorous data to conclude that the spheroids were stem cells in the classical defined sense of the term by stem like is acceptable.

Overall the manuscript is well written contained large amount of novel data and is acceptable for publication at this point.

Reviewer #2 (Remarks to the Author):

The authors have responded admirably to the previous critique with detailed responses to the reviewers' comments, additional experimental data and appropriate revisions to the text. The quality and potential impact of the manuscript is improved as a result.

There are a still a few very minor issues that should be addressed. The authors still need to correct their terminology in a few places: for example, in the diagrams in Figure 4a-c. On line 130, instead of "stem cell-like cells with a stem cell phenotype," revision to "cells with a stem cell-like phenotype" would be more appropriate as the authors' data indicates these cells do not display a "stem cell phenotype." Also, I recommend revising the subheading "HH signaling pathway is involved in disease progression of keloid," to something that indicates that this is a keloid model (e.g., "...involved in disease progression in a keloid model"), not actually keloid disorder. It is important to not over-interpret the data and assume that the model is the same as the human disorder, which the authors have done well in most of the manuscript.

Minor issue: unless the journal requires otherwise, all abbreviations should be defined in each figure legend to minimize reader confusion (there are a lot of abbreviations, and they're all very similar to each other, which can lead to confusion).

Responses to the reviewers' comments

MS: COMMSBIO-23-1553-A

Title: Keloid pathogenesis and fibroproliferative properties are dependent on stem cells regulated by the HEDGEHOG-GLI1 pathway

Authors: Mamiko Tosa, Yoshinori Abe, Seiko Egawa, Tomoka Hatakeyama, Chihiro Iwaguro, Ryotaro Mitsugi, Ayaka Moriyama, Takumi Sano, Rei Ogawa, Nobuyuki Tanaka

We are grateful to the referees for their invaluable comments and suggestions. In accordance with suggestions from the Reviewer #2, we have amended the paper accordingly. Please find below our point-by-point responses to each of the comments.

Reviewer #2 (Remarks to the Author):

The authors have responded admirably to the previous critique with detailed responses to the reviewers' comments, additional experimental data and appropriate revisions to the text. The quality and potential impact of the manuscript is improved as a result.

There are a still a few very minor issues that should be addressed. The authors still need to correct their terminology in a few places: for example, in the diagrams in Figure 4a-c. On line 130, instead of "stem cell-like cells with a stem cell phenotype," revision to "cells with a stem cell-like phenotype" would be more appropriate as the authors' data indicates these cells do not display a "stem cell phenotype."

We agree with this comment and have changed "stem cell-like cells with a stem cell phenotype" to "cells with a stem cell-like phenotype" (lines 129 to 130).

Also, I recommend revising the subheading "HH signaling pathway is involved in disease progression of keloid," to something that indicates that this is a keloid model (e.g., "...involved in disease progression in a keloid model"), not actually keloid disorder. It is

important to not over-interpret the data and assume that the model is the same as the human disorder, which the authors have done well in most of the manuscript.

We also agree with this comment and have changed the subheading to “HH signaling pathway is involved in disease progression of keloid model” (line 224).

Minor issue: unless the journal requires otherwise, all abbreviations should be defined in each figure legend to minimize reader confusion (there are a lot of abbreviations, and they're all very similar to each other, which can lead to confusion).

According to this comment, we added the following descriptions in to the Figure legends as follows. Fig. 1a; Biopsy site from normal skin (non-wounded), keloid center, keloid marginal area, and normal skin adjacent to the keloid from which in vitro primary fibroblasts (transcriptome profiling of normal fibroblasts [NF], fibroblasts from normal-looking skin adjacent keloids [KNF], keloid marginal area [KMF], and keloid central area [KCF], and fibroblast-derived stem-like cells (normal dermis [NS], keloid normal dermis [KNS], keloid marginal area [KMS], and keloid central area [KCS]) were subsequently established (lines 757-762).

Fig. 4 a–c; IPA suggested that GLI1 target secretory molecules (SPP1 and FGF1) from stem-like cells provide site-specific keloid fibroblast function (KNS–KNF [keloid normal] axis: **a**, KMS–KMF [keloid marginal area] axis: **b**, KCS–KCF [keloid central area] axis (lines 817 to 819).

Fig. 5a; Representative image of control and GLI1-knocked down spheres (keloid fibroblast-derived stem like cells) (lines 845 to 846).